# KNOWLEDGE-IN-CONTEXT: TOWARDS KNOWLEDGEABLE SEMI-PARAMETRIC LANGUAGE MODELS

**Xiaoman Pan, Wenlin Yao, Hongming Zhang, Dian Yu, Dong Yu & Jianshu Chen** *
Tencent AI Lab, Bellevue, WA 98004, USA

## ABSTRACT

Fully-parametric language models generally require a huge number of model parameters to store the necessary knowledge for solving multiple natural language tasks in zero/few-shot settings. In addition, it is hard to adapt to the evolving world knowledge without the costly model re-training. In this paper, we develop a novel semi-parametric language model architecture, *Knowledge-in-Context (KiC)*, which empowers a parametric text-to-text language model with a knowledge-rich external memory. Specifically, the external memory contains six different types of knowledge: entity, dictionary, commonsense, event, script, and causality knowledge. For each input instance, the KiC model adaptively selects a knowledge type and retrieves the most helpful pieces of knowledge. The input instance along with its knowledge augmentation is fed into a text-to-text model (e.g., T5) to generate the output answer, where both the input and the output are in natural language forms after prompting. Interestingly, we find that KiC can be identified as a special mixture-of-experts (MoE) model, where the knowledge selector plays the role of a router that is used to determine the sequence-to-expert assignment in MoE. This key observation inspires us to develop a novel algorithm for training KiC with an instance-adaptive knowledge selector. As a knowledge-rich semi-parametric language model, KiC only needs a much smaller parametric part to achieve superior zero-shot performance on unseen tasks. By evaluating on 40+ different tasks, we show that $KiC_{Large}$ with 770M parameters easily outperforms large language models that are 4-39x larger. In addition, KiC also exhibits emergent abilities at a much smaller model scale compared to the fully-parametric models.

## 1 INTRODUCTION

Recently, large-scale fully-parametric language models have achieved great success in solving natural language processing (NLP) tasks (Radford et al., 2019; Brown et al., 2020; Chowdhery et al., 2022; Kaplan et al., 2020). However, they generally require a huge number of model parameters to store the necessary knowledge for solving multiple NLP tasks in the zero/few-shot setting. Meanwhile, their problem solving capability only emerges after reaching a certain model scale (Wei et al., 2022). In addition, large parametric language models are hard to adapt to the evolving world knowledge without expensive model re-training. To overcome these challenges, there has been an increasing interest in developing semi-parametric language models, where a parametric language model is augmented with an external memory containing a large number of text chunks (Borgeaud et al., 2022; Izacard et al., 2022; Khandelwal et al., 2019; Zhong et al., 2022). Although these semi-parametric approaches are shown to be more effective than their much larger parametric counterparts, there remain several challenges. The first challenge is that useful knowledge pieces are generally *sparsely* distributed over a large textual corpus. Therefore, it is difficult to locate and retrieve the correct text chunk that contains the right knowledge to complement a given input instance. Second, it is difficult to determine the proper text chunk granularity to cover the desired knowledge. Thus, people usually use oversized text chunks to build indexing, which makes it even harder to determine whether knowledge is contained. On the other hand, there have been a rich collection of knowledge resources (e.g., knowledge graphs), where different kinds of knowledge are *densely* and *compactly* organized in structured or semi-structured forms. In this paper, we leverage these knowledge resources to construct

---

*{xiaomanpan,wenlinyao,hongmingzhang,yudian,dyu,jianshuchen}@global.tencent.com

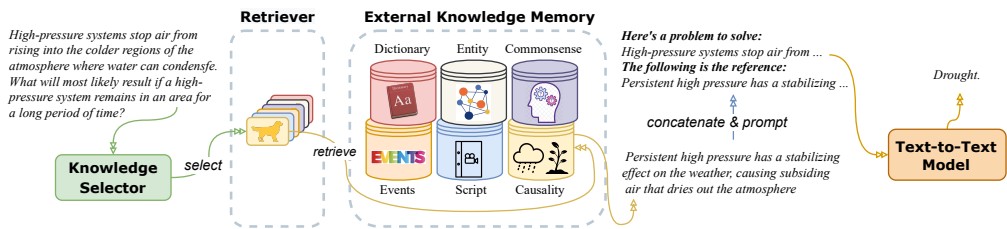

Figure 1: Overview of the KiC model architecture. It is augmented with a knowledge-rich memory that contains diverse categories of knowledge. For each input instance, KiC first selects a particular knowledge category and retrieves the most helpful knowledge pieces to augment the input. It then feeds the prompted input into a text-to-text backbone module (e.g., T5) to generate the output answer.

a semi-parametric language model, by simply using off-shelf encoders and retrievers to index and search the external memory.

In particular, our primary contribution is developing a novel semi-parametric language model architecture, *Knowledge-in-Context (KiC)*, that is fueled by a large *knowledge-rich* external memory (Section 2). Specifically, the memory covers six broad categories of knowledge types: entity, dictionary, commonsense, event, script and causality (Section 2.2). Our comprehensive analysis reveals that a wide range of natural language tasks (31 out of 35 tasks) benefit from adding knowledge, where different knowledge resources help with different subsets of tasks. Interestingly, some tasks are even improved by 10%+ after adding suitable knowledge. To adaptively utilize knowledge, we exploit KiC to dynamically identify the most useful knowledge pieces for each input instance from a certain task and place them in the current context for answering the question. We adopt a single text-to-text transformer (e.g., T5) to generate the output answer from the input. Specifically, we append the retrieved knowledge pieces to the input instance, and then feed them into the text-to-text model to generate the output answer (also in natural language). The major advantage of such a text-to-text paradigm is that it handles multiple natural language tasks with the same interface and can also generalize to unseen tasks (Sanh et al., 2022; Raffel et al., 2020). Moreover, we find this training paradigm is suitable for our model design as it can teach our KiC model to learn how to select and use knowledge through various seen language tasks and then generalize well to use knowledge for solving unseen tasks. Our experimental analysis further shows that such instance-adaptive (context-dependent) knowledge augmentation is critical to the success of KiC model. However, due to the inherent discrete nature, it is difficult to train KiC in a fully differentiable manner to select the correct knowledge category for each instance. To solve this problem, we find that KiC can be reformulated as a special mixture-of-experts (MoE) model (Jacobs et al., 1991; Jordan & Jacobs, 1994; Shazeer et al., 2017; Fedus et al., 2022), where the knowledge selector is identified as the router that is used to determine the sequence-to-expert assignment in MoE (Section 2.3). Furthermore, the memory partition corresponding to each knowledge category together with the text-to-text model can be recognized as a special semi-parametric expert in MoE. This key observation inspires us to develop a novel learning algorithm to train KiC with instance-adaptive knowledge selection capabilities.

In our experiments (Section 3), we adopt the same setting as T0 (Sanh et al., 2022), where we train KiC models on a collection of tasks and then evaluate on another set of unseen tasks in a zero-shot manner. As a knowledge-rich semi-parametric language model, KiC only needs a much smaller parametric part to achieve superior zero-shot performance on unseen tasks. With only 0.77B parameters, KiC_Large outperforms zero-shot baseline models such as GPT-NeoX-20B or OPT-30B that are 25-38x larger. It achieves 39.4% zero-shot performance on MMLU benchmark, very close to the GPT-3's 5-shot performance of 43.9% that has 175B parameters (227x larger). Also, KiC exhibits emergent abilities at a much smaller model scale compared to the fully-parametric models.

## 2 KNOLWEDGE-IN-CONTEXT LANGUAGE MODEL

### 2.1 OVERVIEW

In this section, we introduce our proposed KiC language model, which augments a parametric text-to-text Transformer (backbone) model with a knowledge-rich external memory (Figure 1). Overall, KiC

consists of the following modules: (i) a parametric text-to-text backbone, (ii) an external knowledge memory with a retriever, and (iii) a knowledge selector. As shown in Figure 1, for each input instance, the knowledge selector first selects a particular knowledge category based on the input context and then retrieves the most helpful knowledge pieces for solving the current problem. The retrieved knowledge is used to complement the input context via concatenation, and the knowledge-augmented textual inputs are fed into the text-to-text backbone model, which generates the output solution in natural language. The text-to-text backbone model can be any encoder-decoder models (e.g., T5, BART) or decoder-only models (e.g., GPT, PaLM). For convenience and without loss of generality, we adopt T5 as our backbone model throughout this paper. In the following subsections, we will explain in detail how to construct the knowledge memory along with its retriever (Section 2.2) as well as how to learn the entire KiC model in a fully-differentiable end-to-end manner (Section 2.3).

## 2.2 EXTERNAL KNOWLEDGE MEMORY AND RETRIEVER

**Knowledge-rich external memory** A significant advantage of semi-parametric models over fully-parametric ones is that we could flexibly change the knowledge resources. As shown in Table 7, structured or semi-structured knowledge resources can often provide more relevant and accurate knowledge than plain text. In this work, we include the following popular representative knowledge resources, where each knowledge piece is in the form of < *subject*, *relation*, *object* > triplet. More details about the statistics and examples of these knowledge resources can be found in Appendix A.1.

- **Dictionary:** We consider dictionary (lexical) knowledge, which records definitions and example sentences of English words. We leverage the largest open-source dictionary Wiktionary[1] as the lexical knowledge resource (e.g., < *"apple"*, *definition*, *"A common, round fruit ..."* >). Specifically, we use the Wiktionary dump dated April 30, 2022 that contains 1.3M word definitions and 470K example sentences for 1M words/phrases.

- **Commonsense:** We include commonsense knowledge from ConceptNet (Liu & Singh, 2004), which covers broad knowledge in our daily life. In ConceptNet, all knowledge pieces are represented in the format of triplets with human-defined relations (e.g., < *"bird"*, *CapableOf*, *"fly"* >). We follow previous works (Zhang et al., 2020) to include the core 600K high-quality triplets.

- **Entity:** We cover named entity knowledge in Wikipedia and Wikidata (Vrandečić & Krötzsch, 2014). For each entity (e.g., *United States*), we collect its Wikidata properties (e.g., < *"United States"*, *capital*, *"Washington D.C."* >), and its related Wikipedia sentences (e.g., < *"United States"*, *context*, *"It consists of 50 states ..."* >). Here, related sentences refer to the sentences from an entity's own article, or the sentences of other articles that link to this entity.

- **Event:** We consider knowledge about daily events with human-constructed (i.e., ATOMIC (Hwang et al., 2021) and GLUCOSE (Mostafazadeh et al., 2020)) and auto-extracted event knowledge graphs (i.e., ASER (Zhang et al., 2022a)). Similar to commonsense knowledge, all event knowledge graphs store knowledge in the triplet format, where relations are human-defined or discourse relations, the subject and the object are events (e.g., < *"I am hungry"*, *before*, *"I eat food"* >).

- **Script:** We also include the script knowledge from Sun et al. (2022), which implicitly represents complex relations by situating argument pairs in a context (mostly natural conversations). Specifically, we use 325K triples that are in the form of < *verbal information*, *context*, *nonverbal information* >, where verbal information is an utterance, nonverbal information can be body movements, vocal tones, or facial expressions, etc., and context is the entire text of the scene from which the verbal-nonverbal pair is extracted.

- **Causality**[2]**:** The last external knowledge resource we include is the auto-extracted causal knowledge CausalBank (Li et al., 2020), which collects large-scale English sentences expressing cause-effect relations. It consists of 133M *because*-mode sentences (i.e., sentences captured by 12 patterns such as *"because"*, *"caused by"*, etc.) and 181M *therefore*-mode sentences (i.e., sentences captured by 19 patterns such as *"therefore"*, *"result in"*, etc.). We also convert each sentence into a triplet form (e.g., < *"babies cry"*, *therefore-mode*, *"will lead to sleep problems"* >).

---

[1] https://en.wiktionary.org/wiki/Wiktionary:Main_Page

[2] Follow the literatures in the commonsense community (Zhang et al., 2021; 2022b), we use the term "causality" to refer to commonsense causality, which is mostly contributory (Bunge, 2017).

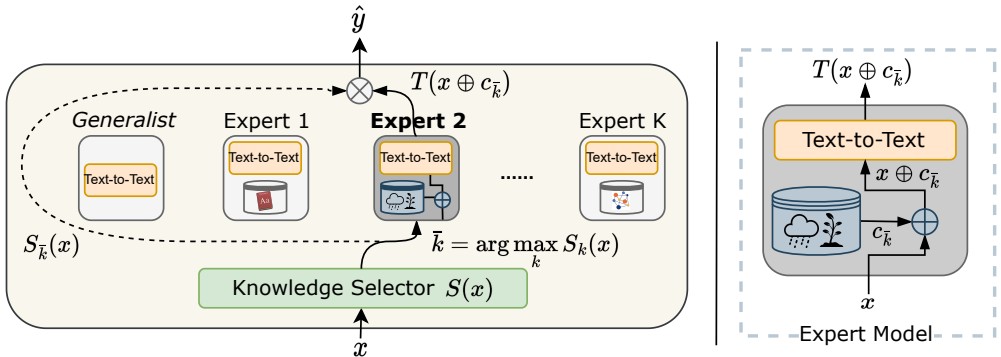

Figure 2: KiC model can be equivalently formulated as a mixture-of-experts (MoE) architecture. The knowledge selector can be identified as a router that is used to determine the sequence-to-expert assignment in MoE. Each expert is made up of the (shared) text-to-text model and the external memory of a particular knowledge category. Therefore, each expert is in itself a stand-alone semi-parametric language model specialized in a certain type of knowledge. To allow the option of not using any knowledge, we also include a "generalist" module, which is the (shared) text-to-text model alone.

Note that although the effectiveness of certain knowledge types such as entity and dictionary knowledge has been demonstrated on a wide range of tasks (e.g., (Zhang et al., 2019b)), other types of knowledge such as commonsense and script knowledge are only used for carefully selected tasks that tend to require these types of knowledge (Ye et al., 2019; Qiu et al., 2019). In this paper, we evaluate the contribution of all aforementioned knowledge types on broader sets of downstream tasks to better understand the contribution of these knowledge types. Some examples of retrieved knowledge can be found in Appendix D, which show their usefulness for solving different tasks.

**Retriever** To effectively retrieve knowledge from the knowledge memory, we follow the previous work (Borgeaud et al., 2022) to use dense retrieval techniques. Specifically, for each knowledge resource, we design one or more knowledge-specific strategies to generate key-value pairs from the original knowledge pieces (see Table 8 in Appendix for details). Then we encode all keys into dense vectors using a SOTA sentence encoder MPNet (Song et al., 2020). During retrieval[3], given a query, we encode it with the same sentence encoder model and then retrieve the most relevant knowledge using the maximum inner product search (MIPS), which is able to reduce search complexity from $O(n)$ to $O(\log n)$. In KiC, we employ SCaNN (Guo et al., 2020) as the MIPS search algorithm.

### 2.3 KIC: A MIXTURE OF SEMI-PARAMETRIC EXPERTS

As we will show in our comprehensive analysis (Table 1), for a particular task, some knowledge categories help the performance while others might hurt. For this reason, it is critical to dynamically select the correct knowledge type in order to facilitate the solution of the problem. In our work, instead of using *task-dependent* knowledge selection, we consider a more fine-grained *instance-dependent* strategy: we adaptively choose the knowledge based on each input instance. We now proceed to explain how KiC learns to make such instance-dependent knowledge selection.

Note that the discrete decision made by the knowledge selector will seep into the overall neural architecture in the form of a discrete latent variable. There could be several alternative methods (such as reinforcement learning (Sutton & Barto, 2018)) for learning the model with discrete latent variables. In this paper, we develop a simple yet effective approach for learning KiC in a fully-differentiable end-to-end manner. The key idea is based on an important observation that KiC can be reformulated as a special one-layer mixture-of-experts architecture, as shown in Figure 2. Note that the knowledge selector can be identified as the router that is used to determine the sequence-to-expert assignment in MoE. This is slightly different from the settings of the recent MoE works (Shazeer et al., 2017; Fedus et al., 2022), where their routers perform token-to-expert assignments. Meanwhile, each expert

---

[3]To further enhance retrieval quality and decrease search space, we employ an additional filtering step for dictionary and entity knowledge pieces. See Appendix A.2 for more knowledge retrieval details.

is made up of the text-to-text module together with a particular category of knowledge memory. Interestingly, each expert is in itself a stand-alone semi-parametric language model, which retrieves a particular kind of knowledge from its own memory to augment its inputs. In other words, each expert can be understood as a *specialist* with expertise in a specific knowledge category. In addition, we also include a special expert named *generalist*, which is used to handle situations where we do not need knowledge from our memory. Furthermore, due to the original KiC design, the text-to-text modules in all the experts (and the generalist) share the same model parameters with the only difference being the non-parametric parts (i.e., the knowledge memories).

Inspired by the above KiC-MoE equivalence, we now proceed to develop a fully-differentiable learning strategy for KiC by leveraging existing MoE learning approaches used in Fedus et al. (2022). More formally, the knowledge selector $S(x)$ is modeled as a $(K + 1)$-class linear classifier, which outputs a $(K + 1)$-dimensional normalized probability vector. We apply the same encoder from our T5 backbone model to the input text sequence from a particular task, which generates a sequence of hidden representation vectors. Then, we apply mean-pooling to them to obtain a fixed-dimension vector, which is fed into the $(K + 1)$-way linear classifier to generate the probabilities of selecting different knowledge categories. Its $k$-th element, denoted as $S_k(x)$, represents the probability of choosing the $k$-th knowledge category for $k = 0, 1, \ldots, K$, where $k = 0$ represents the choice of generalist (i.e., no external knowledge). Let $T(\cdot)$ denote the text-to-text transformer and $c_k$ be the knowledge retrieved from the $k$-th category. Then, in KiC, we select the top-1 knowledge category according to $S(x)$ and compute the output according to the following expressions:

$$\bar{k} = \arg\max_k S_k(x) \tag{1}$$

$$\hat{y} = T(x \oplus c_{\bar{k}}) \cdot S_{\bar{k}}(x) \tag{2}$$

where $\oplus$ denotes concatenation of the input $x$ and the retrieved knowledge $c_{\bar{k}}$ (both in the form of natural language). Observe that KiC first selects the knowledge category $\bar{k}$ that has the highest probability, and then retrieves the most relevant knowledge $c_{\bar{k}}$ from that category to complement the input $x$. The knowledge-augmented input is fed into the text-to-text model to generate the logits for the output tokens. Similar to SwitchTransformer (Fedus et al., 2022), we multiply the output logits from $T(\cdot)$ by the probability $S_{\bar{k}}(x)$ from the selector to compute the final logits for the output tokens. This is a simple yet quite effective strategy to enable differentiable learning in MoE, which was successfully used in both Shazeer et al. (2017) and Fedus et al. (2022). We adopt this similar strategy and our experiments in Section 3 will demonstrate its effectiveness in KiC learning as well.[4] Note that we currently only consider the top-1 knowledge selection (routing) for simplicity and leave the generalization to top-n selection as future work. Finally, similar to MoE, we also add an auxiliary load balancing loss together with the standard cross-entropy loss during KiC learning:

$$\mathcal{L}(x, y) = \sum_{t=1}^{T} \texttt{CrossEntropy}(\hat{y}_t, y_t) + \alpha \cdot \texttt{Balancing}(S(x)) \tag{3}$$

where $y$ denotes the target sequence, the subscript $t$ indexes the $t$-th output token, and $\alpha$ is a positive hyper-parameter that controls the tradeoff between the two losses. We find that, without a load balancing term, the knowledge selector tends to select only one knowledge category throughout the entire training process, which was also observed in MoE learning. There could be different choices of the balancing loss such as the ones used in (Shazeer et al., 2017; Fedus et al., 2022), which encourage the diversity of knowledge selection in different ways based on $S(x)$. Without loss of generality, we use the same load balancing loss as in SwitchTransformer (Fedus et al., 2022) (see Equation 4).

The above KiC-MoE equivalence may also lead to interesting observations that could potentially benefit the studies of both semi-parametric language models and MoEs. For example, in MoE works, the experts are generally designed to be different parametric neural modules (e.g., different MLPs (Fedus et al., 2022; Shazeer et al., 2017)). However, our work shows that this may not be the only option: we can construct different experts by using the same parametric module but with different inputs. By bridging these two active areas, we hope there could be more fruitful future outcomes.

---

[4]It might be tempting to use Gumbel-Softmax to handle the discrete latent variable in KiC. However, in order to use the straight-through-estimator during backpropagation, it has to compute the hidden states for all the experts, i.e., executing the text-to-text transformer by $(K + 1)$ times, which is prohibitive when $K$ increases.

## 3 EXPERIMENTS

### 3.1 ANALYSIS OF KNOWLEDGE USEFULNESS

To verify our assumption that external knowledge resources can facilitate LMs in general language understanding and see the effects of using different types of knowledge, we conduct single-task fine-tuning experiments on a wide range of downstream tasks (Table 1). We evaluate 35 tasks in total and classify them into 10 categories following the P3 task categorization framework (Sanh et al., 2022). For each knowledge type (each column), we append retrieved knowledge pieces to the input sentence and truncate the entire sequence whenever it exceeds the sequence limit. Next, the augmented input sentences are fed into the standard text-to-text model (T5) to generate the target answer for optimization, where training instances are from every single task. We can see that model performances on 30 out of 35 tasks are improved after adding at least one type of knowledge, which demonstrates the effectiveness of using high-quality external knowledge. Based on these results, we exploit KiC to dynamically identify the most useful knowledge pieces to adaptively utilize knowledge.

| Category | Task | Task Reference | None | ENT | DIC | COM | EVT | SCR | CAU |
|---|---|---|---|---|---|---|---|---|---|
| Coreference | WSC | Levesque et al. (2012) | 60.3 | 49.5 | 62.0 | 53.1 | **64.7** | 62.0 | 63.4 |
| | Wino. debiased | Sakaguchi et al. (2021) | **59.1** | 53.5 | 57.3 | 56.9 | 58.3 | 58.5 | 54.1 |
| | Wino. xl | Sakaguchi et al. (2021) | 63.5 | 63.0 | 64.2 | **64.5** | 64.3 | 63.8 | 63.5 |
| NLI | CB | De Marneffe et al. (2019) | 87.5 | 87.5 | 85.9 | 87.5 | 85.9 | **90.6** | 84.4 |
| | RTE | Wang et al. (2019) | 77.1 | 76.2 | 79.0 | **79.2** | 76.6 | 76.9 | 71.3 |
| Paraphrase | MRPC | Dolan & Brockett (2005) | 82.9 | 80.5 | **87.7** | 77.9 | 84.9 | 84.4 | 82.0 |
| | QQP | Wang et al. (2018) | 89.4 | 89.1 | **89.5** | 89.5 | 89.2 | 89.3 | 89.4 |
| | PAWS | Zhang et al. (2019a) | **94.6** | 94.2 | 94.3 | 94.4 | 94.4 | 94.5 | 94.2 |
| Closed QA | ARC-Easy | Clark et al. (2018) | 52.8 | 52.6 | 53.1 | 51.7 | 56.1 | 51.7 | **64.6** |
| | ARC-Challenge | Clark et al. (2018) | 30.9 | 36.2 | 30.9 | 33.5 | 34.2 | 37.2 | **39.5** |
| | WikiQA | Yang et al. (2015) | **96.2** | 95.6 | 95.8 | 95.9 | 95.7 | 95.7 | 96.2 |
| Extr. QA | ReCoRD | Zhang et al. (2018) | 53.9 | 53.9 | 53.2 | 54.0 | **54.1** | 53.9 | 53.5 |
| Multi QA | CoS-E v1.11 | Rajani et al. (2019) | 60.6 | **61.2** | 59.9 | 60.8 | 60.1 | 59.7 | 61.1 |
| | CosmosQA | Huang et al. (2019) | 69.1 | 69.0 | 68.3 | **69.7** | 67.9 | 67.7 | 66.4 |
| | DREAM | Sun et al. (2019) | 62.4 | 63.8 | 63.5 | 62.5 | 63.3 | **63.8** | 62.7 |
| | OpenBookQA | Mihaylov et al. (2018) | 56.2 | 54.7 | 54.7 | 57.4 | **58.2** | 55.7 | 57.6 |
| | PIQA | Bisk et al. (2020) | 71.7 | 72.5 | 71.6 | 71.5 | 71.5 | 70.6 | **74.3** |
| | QASC | Khot et al. (2020) | 97.8 | **98.1** | 98.0 | 98.0 | 98.1 | 97.8 | 97.6 |
| | QuAIL | Rogers et al. (2020) | 68.3 | 68.3 | 72.9 | **73.5** | 72.9 | 66.6 | 68.6 |
| | QuaRTz | Tafjord et al. (2019) | **83.1** | 81.8 | 81.1 | 81.1 | 81.5 | 82.2 | 81.1 |
| | RACE-Middle | Lai et al. (2017) | 74.1 | 73.8 | 74.1 | **74.4** | 73.3 | 73.3 | 72.7 |
| | RACE-High | Lai et al. (2017) | 69.4 | 69.3 | **69.9** | 69.7 | 69.2 | 68.8 | 69.7 |
| | SciQ | Welbl et al. (2017) | 94.0 | 95.5 | 95.0 | 98.0 | 96.6 | 94.1 | **98.7** |
| | SocialIQA | Sap et al. (2019) | 63.4 | 63.5 | 63.6 | **64.2** | 63.7 | 63.9 | 63.2 |
| | BoolQ | Clark et al. (2019) | 81.9 | **82.2** | 81.7 | 82.0 | 80.6 | 81.5 | 81.7 |
| | MultiRC | Khashabi et al. (2018) | 80.0 | 79.7 | 79.5 | **80.0** | 79.5 | 79.2 | 79.0 |
| | WikiHop | Welbl et al. (2018) | 58.7 | 58.7 | 58.8 | **59.4** | 59.4 | 59.1 | 58.5 |
| | WIQA | Tandon et al. (2019) | 74.4 | 74.4 | 75.1 | **83.9** | 83.6 | 82.4 | 83.5 |
| Sentiment | IMDB | Maas et al. (2011) | 94.8 | **94.9** | 94.7 | 94.9 | 94.7 | 94.7 | 94.8 |
| | Rotten Tomatoes | Pang & Lee (2005) | 90.2 | 89.6 | **90.3** | 89.9 | 90.0 | 90.0 | 89.6 |
| Completion | HellaSwag | Zellers et al. (2019) | 49.8 | 49.3 | 50.6 | 51.8 | 52.0 | 49.8 | **53.7** |
| | COPA | Roemmele et al. (2011) | 58.0 | 58.5 | 59.8 | 54.5 | 58.9 | 56.2 | **62.0** |
| Topic Class. | AG News | Del Corso et al. (2005) | 93.9 | 93.6 | 94.0 | **94.3** | 94.0 | 94.1 | 94.1 |
| | DBpedia14 | Lehmann et al. (2015) | **28.4** | 28.4 | 28.4 | 28.4 | 28.4 | 28.4 | 28.4 |
| WSD | WiC | Pilehvar et al. (2019) | 68.8 | 67.9 | 69.5 | **70.2** | 68.2 | 66.3 | 69.8 |

Table 1: Single task fine-tuning results (accuracy %) of using no knowledge (None) or adding entity (ENT), dictionary (DIC), commonsense (COM), event (EVT), script (SCR), or causality (CAU) knowledge separately. For each row, we use green and red to indicate performance increase or decrease in comparison with no knowledge (None). The boldface numbers are the best performance for each row. Note that all results in this table are based on T5$_{\text{Base}}$. Appendix C contains the full description of all tasks.

## 3.2 MAIN RESULTS

| Models | Params | Coreference | | NLI | | | | | Completion | | | WSD |
|---|---|---|---|---|---|---|---|---|---|---|---|---|
| | | WSC | Wino. | $ANLI_{R1}$ | $ANLI_{R2}$ | $ANLI_{R3}$ | CB | RTE | COPA | H.S. | S.C. | WiC |
| BERT | 0.34B | $53.4_{7.8}$ | $49.5_{1.0}$ | $33.8_{1.4}$ | $33.8_{0.9}$ | $33.2_{0.6}$ | $48.2_{11.3}$ | $48.4_{1.7}$ | $49.0_{3.1}$ | $25.5_{0.3}$ | $50.1_{0.1}$ | $50.2_{0.4}$ |
| RoBERTa | 0.35B | $37.0_{3.3}$ | $48.9_{0.7}$ | $33.4_{0.9}$ | $33.4_{0.6}$ | $33.3_{0.5}$ | $42.9_{4.1}$ | $54.1_{1.1}$ | $56.0_{2.3}$ | $22.4_{1.1}$ | $48.5_{0.4}$ | $50.0_{1.0}$ |
| GPT-Neo | 2.7B | $45.2_{8.3}$ | $50.8_{0.9}$ | $33.5_{1.1}$ | $33.3_{0.7}$ | $33.4_{0.5}$ | $48.2_{18.9}$ | $51.1_{3.2}$ | $50.5_{6.3}$ | $25.0_{0.4}$ | $54.6_{1.4}$ | $52.5_{1.3}$ |
| GPT-J | 6B | $40.4_{9.7}$ | $48.5_{0.8}$ | $34.0_{1.2}$ | $33.7_{0.9}$ | $33.6_{0.7}$ | $28.6_{15.0}$ | $50.5_{3.2}$ | $56.0_{4.2}$ | $24.7_{0.5}$ | $53.3_{1.1}$ | $51.0_{3.0}$ |
| OPT | 30B | $63.5_{13.9}$ | $48.4_{0.3}$ | $33.3_{0.2}$ | $33.3_{0.1}$ | $33.4_{0.1}$ | $50.0_{16.7}$ | $47.3_{1.7}$ | $52.0_{1.9}$ | $24.4_{0.4}$ | $55.3_{0.3}$ | $50.0_{0.0}$ |
| GPT-NeoX | 20B | $60.6_{9.4}$ | $48.9_{0.8}$ | $33.4_{1.2}$ | $33.4_{1.0}$ | $33.6_{0.7}$ | $30.4_{14.6}$ | $48.4_{2.7}$ | $44.5_{5.8}$ | $25.0_{0.4}$ | $53.5_{1.0}$ | $49.9_{2.5}$ |
| $T0_{Base}$ | 0.22B | $61.1_{5.5}$ | $50.6_{0.9}$ | $32.2_{1.4}$ | $33.0_{1.0}$ | $34.2_{0.7}$ | $53.6_{17.1}$ | $64.1_{1.4}$ | $65.7_{5.7}$ | $25.7_{0.5}$ | $80.6_{1.3}$ | $50.7_{1.4}$ |
| $T0_{Large}$ | 0.77B | $59.1_{5.9}$ | $50.5_{0.3}$ | $30.5_{2.0}$ | $32.7_{0.6}$ | $33.8_{0.8}$ | $60.7_{23.0}$ | $62.1_{3.1}$ | $73.5_{8.4}$ | $25.7_{0.4}$ | $84.1_{1.8}$ | $50.2_{1.0}$ |
| $T0_{3B}$ | 3B | $64.4_{2.7}$ | $50.5_{1.2}$ | $33.7_{0.9}$ | $33.4_{1.2}$ | $33.3_{0.4}$ | $50.0_{15.9}$ | $64.1_{3.5}$ | $74.9_{8.7}$ | $27.5_{1.0}$ | $85.1_{3.2}$ | $50.4_{0.9}$ |
| $T0_{11B}$ | 11B | $64.4_{6.3}$ | $60.5_{2.5}$ | $44.7_{3.6}$ | $39.4_{2.2}$ | $42.4_{3.0}$ | $78.6_{18.5}$ | $81.2_{3.7}$ | $90.8_{4.1}$ | $33.7_{0.5}$ | $94.7_{4.7}$ | $57.2_{1.8}$ |
| $KiC_{Small}$ | 0.06B | $63.5_{3.9}$ | $51.1_{0.6}$ | $33.3_{1.0}$ | $33.3_{0.9}$ | $33.6_{0.6}$ | $44.6_{12.1}$ | $47.3_{2.4}$ | $48.0_{5.2}$ | $25.4_{0.5}$ | $57.7_{1.7}$ | $50.0_{0.5}$ |
| $KiC_{Base}$ | 0.22B | $63.5_{1.0}$ | $50.0_{0.4}$ | $28.4_{2.4}$ | $30.9_{1.7}$ | $32.8_{1.1}$ | $58.9_{17.2}$ | $66.8_{2.9}$ | $65.0_{9.0}$ | $26.1_{0.7}$ | $82.6_{0.8}$ | $50.2_{1.5}$ |
| $KiC_{Large}$ | 0.77B | $65.4_{8.3}$ | $55.3_{2.4}$ | $36.3_{1.8}$ | $35.0_{1.4}$ | $37.6_{2.5}$ | $67.9_{22.9}$ | $74.0_{3.8}$ | $85.3_{6.8}$ | $29.6_{0.9}$ | $94.4_{1.2}$ | $52.4_{1.5}$ |

Table 2: Zero-shot evaluation results on held-out unseen tasks (Wino.: Winogrande XL; H.S.: HellaSwag; S.C.: StoryCloze). Following previous papers, we report the median accuracy (%) and the standard deviation of all prompts used. Note that $T0_{Base}$ and $T0_{Large}$ are reproduced using the same collection of tasks and hyper-parameters with KiC models. Baseline models are: BERT(Devlin et al., 2019), RoBERTa(Liu et al., 2019), GPT-Neo(Black et al., 2021), GPT-J(Wang & Komatsuzaki, 2021), GPT-NeoX(Black et al., 2022), OPT(Zhang et al., 2022c). We use the standard autoregressive (log) probabilities to score candidate choices and select the best one as the prediction for all baseline models including mask LMs such as BERT and RoBERTa.

| Models | Params | Method | STEM | Humanities | Social Science | Other | Average |
|---|---|---|---|---|---|---|---|
| $RoBERTa_{Large}$ | 0.35B | fine-tune | 27.0 | 27.9 | 28.8 | 27.7 | 27.9 |
| GPT-2 | 1.5B | fine-tune | 30.2 | 32.8 | 33.3 | 33.1 | 32.4 |
| Gopher | 7.1B | 5-shot | 30.1 | 28.0 | 31.0 | 31.0 | 29.5 |
| Atlas | 11B | 5-shot | 38.8 | 46.1 | 54.6 | 52.8 | 47.9 |
| GPT-3 | 13B | 5-shot | 24.3 | 27.1 | 25.6 | 26.5 | 26.0 |
| GPT-NeoX | 20B | 5-shot | 34.9 | 29.8 | 33.7 | 37.7 | 33.6 |
| GPT-3 | 175B | 5-shot | 36.7 | 40.8 | 50.4 | 48.8 | 43.9 |
| GPT-Neo | 2.7B | 0-shot | 28.2 | 30.1 | 21.9 | 24.4 | 26.1 |
| $T0_{3B}$ | 3B | 0-shot | 29.9 | 34.2 | 40.4 | 38.1 | 35.7 |
| GPT-J | 6B | 0-shot | 26.9 | 29.3 | 29.2 | 27.4 | 28.2 |
| $T0_{11B}$ | 11B | 0-shot | 33.3 | 42.2 | 48.5 | 48.9 | 43.2 |
| Atlas | 11B | 0-shot | 38.0 | 43.6 | 54.1 | 54.4 | 47.5 |
| GPT-NeoX | 20B | 0-shot | 29.2 | 29.9 | 28.5 | 27.0 | 28.7 |
| OPT | 30B | 0-shot | 27.7 | 30.1 | 27.0 | 28.6 | 28.4 |
| $KiC_{Small}$ | 0.06B | 0-shot | 26.4 | 26.6 | 26.8 | 27.5 | 26.8 |
| $KiC_{Base}$ | 0.22B | 0-shot | 27.9 | 30.7 | 33.4 | 33.7 | 31.4 |
| $KiC_{Large}$ | 0.77B | 0-shot | 30.7 | 38.3 | 43.6 | 44.8 | 39.4 |

Table 3: Comparison to state-of-the-art results on the test set of MMLU tasks. Following standard approaches, we choose the prompt that yields the best accuracy (%) on the validation set. Additional models used for comparison: Gopher (Rae et al., 2021), Atlas (Izacard et al., 2022).

Our main model KiC is initialized with $T5_{LM-adapt}$, an improved version of T5 that continues training T5 for additional 100K steps on the LM objective (Lester et al., 2021) to enhance its ability to generate natural language. Similar to T0, we train our KiC model on a mixture of multiple tasks (39 tasks in total) by combining and shuffling all training instances from different tasks (8.4M in total) and predict on unseen (held-out) tasks to evaluate zero-shot generalization ability. Our final $KiC_{Large}$ model is trained with 128 V100 GPUs for 42 hours. More training details are in Appendix A.2.

**Zero-shot generalization** We evaluate our KiC model on two groups of zero-shot datasets. 1) Held-out tasks of P3 contain two coreference tasks, three NLI tasks, three sentence completion tasks and one word sense disambiguation (WSD) task. Table 2 shows that our $KiC_{Large}$ model outperforms

| Model | Paraphrase | | | Close Book QA | | | Multi-Choice QA | | |
|---|---|---|---|---|---|---|---|---|---|
| | MRPC | QQP | PAWS | ARC$_{Easy}$* | ARC$_{Challenge}$* | WikiQA | CoS-E$_{v1.11}$ | CosmosQA | SciQ |
| T0$_{Large}$ | $85.3_{0.8}$ | $88.6_{0.1}$ | $94.7_{0.0}$ | $50.9_{1.4}$ | $37.5_{0.5}$ | $95.4_{0.1}$ | $58.1_{0.3}$ | $71.2_{23.9}$ | $92.5_{13.4}$ |
| KiC$_{Large}$ | $85.5_{0.7}$ | $89.3_{0.3}$ | $95.3_{0.1}$ | $67.9_{1.6}$ | $46.5_{1.1}$ | $95.7_{0.3}$ | $76.3_{0.2}$ | $81.3_{28.3}$ | $94.6_{11.0}$ |

| Model | Multi-Choice QA | | | | | | | |
|---|---|---|---|---|---|---|---|---|
| | DREAM | OpenBookQA* | PIQA* | QASC | QuAIL | QuaRTz | RACE$_{Middle}$* | RACE$_{High}$* |
| T0$_{Large}$ | $72.9_{0.0}$ | $38.8_{1.9}$ | $53.2_{2.1}$ | $97.6_{4.0}$ | $71.4_{17.3}$ | $84.8_{0.8}$ | $61.6_{6.1}$ | $49.6_{7.3}$ |
| KiC$_{Large}$ | $82.5_{0.1}$ | $53.4_{4.9}$ | $64.6_{7.9}$ | $99.1_{4.2}$ | $78.9_{19.9}$ | $89.7_{0.8}$ | $74.8_{11.0}$ | $65.7_{9.1}$ |

| Model | Multi-Choice QA | | | | Sentiment Analysis | | Topic Classification | |
|---|---|---|---|---|---|---|---|---|
| | SocialIQA | BoolQ* | WikiHop | WIQA | IMDB† | Rotten Tomatoes | AG News*† | DBpedia14*† |
| T0$_{Large}$ | $67.7_{10.0}$ | $62.6_{1.2}$ | $60.9_{0.1}$ | $75.8_{8.3}$ | $95.2_{27.4}$ | $87.8_{0.3}$ | $94.0_{0.0}$ | $28.4_{2.9}$ |
| KiC$_{Large}$ | $74.2_{11.5}$ | $72.9_{2.3}$ | $58.8_{0.1}$ | $79.1_{8.5}$ | $96.5_{27.7}$ | $91.7_{0.2}$ | $94.2_{0.1}$ | $31.3_{5.3}$ |

Table 4: In-domain evaluation results measured in accuracy (%) and standard deviation. T0$_{Large}$ and KiC$_{Large}$ are trained using the same collection of tasks and hyper-parameters, while KiC$_{Large}$ has the knowledge selector during multitask learning. * indicates that the training data provided by this task are not used in multitask training. Thus, we regard tasks with * as in-domain zero-shot evaluation because KiC has observed similar tasks (such as other multi-choice QA tasks) in multitask training. † indicates that it's the score on the test set. Otherwise, we report the score on the validation set.

all zeroshot baseline models (e.g., GPT-NeoX, OPT) that are 25-38x larger. Moreover, KiC$_{Large}$ beats T0$_{3B}$ that has 3B parameters on all 9 tasks by a large margin with our adaptive knowledge selector and only 0.77B parameters. 2) Massive Multitask Language Understanding (MMLU) (Hendrycks et al., 2020) benchmark is designed to measure knowledge acquired in model pretraining. MMLU covers 57 subjects under four categories, i.e., STEM, Humanities, Social Sciences and Other. Comparisons with SOTA LMs are shown in Table 3. We can see that KiC$_{Large}$ beats all fine-tuning baseline models RoBERTa$_{Large}$ and GPT-2 without using any training data from MMLU. Surprisingly, KiC$_{Large}$ achieves an average performance of 39.4% using only 0.77B parameters, which is just 4.5% below the 5-shot performance of GPT-3 that has 175B parameters (227x larger). To investigate how the KiC knowledge selector leverages different knowledge resources when applying to unseen tasks, we plot the distributions of the selected knowledge categories in Figure 4. More discussions and analysis can be found in Appendix B. Finally, to examine the importance of different KiC components (e.g., knowledge selectors, external knowledge sources, etc.), we conduct extensive ablation studies by comparing our full KiC model with the following baselines: (i) KiC without knowledge, (ii) KiC with an external memory that contains only plain text (English Wikipedia), (iii) KiC without knowledge-selector but retrieving from a mixture of all knowledge categories, (iv) KiC with a task-adaptive selector, and (v) KiC without generalist. The results are reported in Table 12 of Appendix B.

**KiC in multi-task training** To see whether our KiC learning can help with multi-tasking training, we reproduce T0$_{Large}$ with the same collection of tasks and evaluate KiC$_{Large}$ on the validation set of each in-domain task (Table 4). Here, in-domain tasks can be divided into two groups - tasks used in multitask training and tasks not used in multitask training but within the observed task category. Again, KiC$_{Large}$ outperforms T0$_{Large}$, with significant improvement on in-domain unseen tasks (tasks marked with *) such as Race and BoolQ and knowledge-intensive tasks such as CosmosQA and DREAM. It demonstrates the superiority of our proposed KiC learning in multi-tasking training.

**Emerging behavior** Wei et al. (2022) discover that language models usually can only perform a near-random zero/few-shot performance when they are small but achieves a substantial performance jump when they reach a certain critical threshold of scale (size). A language model is generally considered superior if it can show emerging behavior at a smaller model scale. Therefore, we compare our KiC model with T5 and T0 on held-out tasks to see how performance change with respect to their model sizes. From Figure 3, we can see that T5 is around random guess when the model is below 11B. T0 is better than T5 as it shows emerging behavior when it increases from 3B to 11B. Surprisingly, our KiC model shows emerging behavior when it increases from 0.22B to 0.77B, which demonstrates that our semi-parametric model can achieve the same language understanding capacity using much fewer parameters with the help of adaptive knowledge selector and external knowledge.

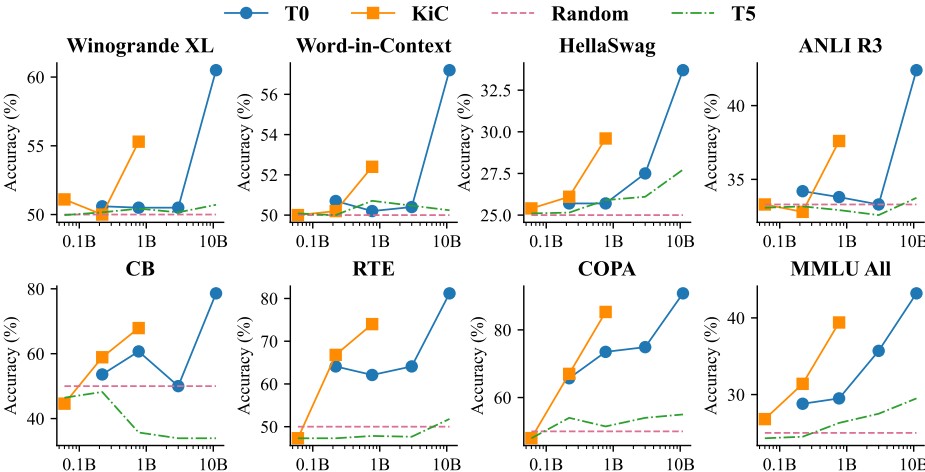

Figure 3: Emerging behaviors of T5, T0 and KiC models. Our KiC model shows emerging behavior at a much smaller model scale (when it increases from 0.22B to 0.77B) compared to T0.

## 4 RELATED WORK

**Knowledge Injection of PLMs**    Although PLMs can capture knowledge such as linguistic, semantic, commonsense, and world knowledge to some extent, they can only memorize knowledge vaguely in parameters, causing poor performance on knowledge-intensive tasks. Recent studies make a great effort to inject knowledge such as lexical knowledge, entity knowledge graph, or syntactic knowledge into LM pre-training (Yang et al., 2021). For example, besides masked language modeling (MLM) and next sentence prediction (NSP), Lauscher et al. (2020) add synonyms and hyponym-hypernym relation prediction between words and Levine et al. (2020) add supersense prediction of masked words into LM training objectives. To use entity knowledge, ERNIE 2.0 (Sun et al., 2020) introduces named entity masking to learn better embeddings for semantic units, Peters et al. (2019) include entity linking, hypernym linking into pre-training and K-BERT (Liu et al., 2020) uses entity knowledge triples to construct knowledge-rich sentence trees. For syntax knowledge injection, Wang et al. (2021) integrate dependency relation prediction into LM training and Bai et al. (2021) incorporate syntax tree information through a syntax-aware self-attention mechanism.

**Semi-parametric language models**    Most of the existing works on semi-parametric language models (Khandelwal et al., 2019; Zhong et al., 2022; Grave et al., 2017; Merity et al., 2017; de Masson d'Autume et al., 2019; Guu et al., 2020; Fan et al., 2021; Lewis et al., 2020) mainly focus on improving the language modeling capability (e.g., improving perplexities) or a particular category of downstream task (e.g., open-domain question answering). Some recent works (Izacard et al., 2022; Borgeaud et al., 2022; Petroni et al., 2021) seek to improve diverse downstream tasks with an external memory. All these works augment the parametric language model with memories of plain texts. In contrast, we focus on developing semi-parametric language models with a knowledge-rich memory for improving the performance of a wide range of downstream language tasks.

## 5 CONCLUSIONS AND FUTURE WORK

This work develops a novel semi-parametric language model architecture, *Knowledge-in-Context (KiC)*, which empowers a parametric text-to-text language model with a knowledge-rich external memory containing six different types of knowledge. We also design an instance-adaptive knowledge selector to retrieve the most helpful pieces of knowledge for each input instance. As a knowledge-rich semi-parametric language model, KiC only needs a relatively smaller parametric part to achieve superior zero-shot performance on unseen tasks and exhibits emergent abilities at a much smaller model scale compared to the fully-parametric models. Future work may include future exploiting unstructured plain texts to pre-train KiC.

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

APPENDIX

## A   EXPERIMENTAL DETAILS

### A.1   KNOWLEDGE PIECES

In this section, we give the basic statistics of different knowledge categories that are used in KiC
— see Table 5. In addition, we further give examples of the knowledge pieces for each category
(Table 6). The knowledge pieces are in the form of < subject, relation, object >. They will be further
encoded into key-value pairs according to different strategies in Appendix A.2.

| | Dictionary | Commonsense | Entity | Event | Script | Causality |
|---|---|---|---|---|---|---|
| # instances | 1.8M | 600K | 257M | 6.4M | 248K | 314M |
| storage | 257MB | 213MB | 155GB | 930MB | 361MB | 36GB |
| type | human | human | human | auto | auto | auto |

Table 5: The statistics of different knowledge categories ("K": thousand, "M": million). Storage is
the space required to store the original data ("MB": megabyte, "GB": gigabyte). The type marked as
"human" means that it is collected by crowd-sourcing, and "auto" means it is automatically extracted.

| | subject | relation | object |
|---|---|---|---|
| Dictionary | *apple* 
 *apple* | *definition* 
 *context* | *A common, round fruit ...* 
 *Apples were washed, then tipped, ...* |
| Commonsense | *bird* 
 *bike* | *CapableOf* 
 *UsedFor* | *fly* 
 *ride* |
| Entity | *United States* 
 *United States* | *capital* 
 *context* | *Washington D.C.* 
 *It consists of 50 states ...* |
| Event | *I am hungry* 
 *I am hungry* | *before* 
 *Conjunction* | *I eat food* 
 *I am tired* |
| Script | *VOICE: Don't leave me.* 






 *DEREK: Michael, my brother, peace* | *MILLER: Peters, do you read me.* 
 *A MAN'S VOICE, in agony, CRACKLES over* 
 *Miller's radio:* 
 *VOICE: Don't leave me.* 
 *MILLER: Justin? Justin, sound off. Justin!"* 
 *Miller trails off as RED LIGHT flickers across* 
 *his visor. He turns.* 
 *Michael waves to DEREK, the one with the* 
 *longest dreads.* 
 *MICHAEL (continuing.): Derek - save some* 
 *for after lunch, bub?* 
 *DEREK: Michael, my brother, peace* 
 *Cameron turns to follow Michael as they walk* 
 *into the cafeteria.* | *radio* 






 *very stoned* |
| Causality | *babies cry* 
 *babies cry* | *therefore-mode* 
 *because-mode* | *will lead to sleep problems* 
 *because they are hungry* |

Table 6: Examples of knowledge piece in the format of <subject, relation, object> triplets. For script
knowledge, < subject, relation, object > becomes < verbal information, context, nonverbal information
> extracted from movie scripts (Sun et al., 2022), where verbal information is an utterance, nonverbal
information can be body movements, vocal tones, or facial expressions, etc., and context is the
entire text of the scene from which the verbal-nonverbal pair is extracted. The verbal and nonverbal
messages are conveyed within a short time period (usually mentioned in the same turn or adjacent
turns). Note that the script knowledge can be viewed as a special kind of commonsense knowledge,
where the relations are characterized by free texts.

### A.2   IMPLEMENTATION DETAILS

**Key-Value pairs construction**   Our knowledge memory consists of a large set of key-value pairs,
which are constructed in the following manner. First, we build an initial set of key-value pairs (in
textual form) from the original knowledge pieces (i.e., knowledge triplets) according to Table 8.
Then, we further encode the keys into dense vectors using MPNet. The encoded keys along with their
corresponding values (in textual forms) will be stored as the final key-value pairs in our knowledge
memory. The encoded key vectors are used for knowledge piece retrieval during MIPS search.

| Question | *High-pressure systems stop air from rising into the colder regions of the atmosphere where water can condense. What will most likely result if a high-pressure system remains in an area for a long period of time?* |
| --- | --- |
| Answer | *Drought* |
| CausalBank (structured) | *Persistent high pressure has a stabilizing effect on the weather, causing subsiding air that dries out the atmosphere.* |
| Wikipedia (plain text) | *High-pressure systems are alternatively referred to as anticyclones. On English - language weather maps, high-pressure centers are identified by the letter H in English, within the isobar with the highest pressure value. On constant pressure upper level charts, it is located within the highest height line contour.* |

Table 7: Examples of retrieved supporting knowledge from different resources (i.e., CausalBank v.s. Wikipedia). Note that the retrieved knowledge pieces from CausalBank are generally more helpful in solving the problem than the retrieved plain text pieces from Wikipedia.

| | Key | Value |
| --- | --- | --- |
| Dictionary | $s$ | $o$ |
| Commonsense | $s$ | $s \oplus r \oplus o$ |
| | $s \oplus o$ | $s \oplus r \oplus o$ |
| | $s \oplus r \oplus o$ | $s \oplus r \oplus o$ |
| Entity | $s$ | $o$ |
| | $o$ | $o$ |
| Event | $s$ | $s \oplus r \oplus o$ |
| | $s \oplus o$ | $s \oplus r \oplus o$ |
| | $s \oplus r \oplus o$ | $s \oplus r \oplus o$ |
| Script | $s$ | $r$ |
| | $o$ | $r$ |
| Causality | $s \oplus o$ | $s \oplus o$ |
| | $o \oplus s$ | $o \oplus s$ |

Table 8: Knowledge-specific strategies to construct key-value pairs from knowledge triplets < *subject* ($s$), *relation* ($r$), *object* ($o$) > ($\oplus$ denotes concatenation). The keys will be further encoded into vector forms using MPNet, which are used for knowledge retrieval during MIPS search.

**Retriever**  We use All-MPNet$_{\text{base-v2}}$[5] as the encoder for encoding the keys in knowledge memory as well as the input query instance. The model is trained on one billion sentence pairs with the contrastive learning objective, and we use the publically available model checkpoint. For most knowledge categories, we directly apply MIPS search to the encoded query and key vectors during retrieval. For the dictionary knowledge and the entity knowledge, we first pre-filter the knowledge pieces according to the following strategies before applying MIPS search.

- When retrieving from dictionary knowledge, we first use a domain-independent keyword extraction algorithm (Rose et al., 2010) to extract important words from the query[6]. Then, we filter the knowledge pieces so that only the ones related to the important words are retained for MIPS search.

- When retrieving from entity knowledge, we follow previous work (Pan et al., 2019) to first extract concept mentions from the query and then link each mention to its corresponding page in Wikipedia. All the knowledge pieces that are not related to the linked concepts are excluded from MIPS search.

The above pre-filtering strategies are also common practices when using these types of knowledge, which allow us to locate relevant knowledge pieces more accurately. In addition, they also reduce the MIPS search complexity by focusing only on the most relevant candidates.

**Load Balancing Loss**  To encourage the diversity of knowledge selection, we adopt the load balancing loss from SwitchTransformer (Fedus et al., 2022). Given $K + 1$ experts, a batch $\mathcal{B}$ with $B$

---

[5]https://huggingface.co/sentence-transformers/all-mpnet-base-v2
[6]https://pypi.org/project/rake-nltk/

sequences, the load balancing loss is computed according to:

$$\texttt{Balancing}\big(S(x)\big) = (K+1) \cdot \sum_{i=0}^{K+1} f_i \cdot P_i, \tag{4}$$

where $f_i$ is the fraction of sequences that are actually dispatched to expert $i$, and $P_i$ is the fraction of the selector probability allocated for expert $i$, which are defined as

$$f_i = \frac{1}{B} \sum_{x \in \mathcal{B}} \mathbb{1}\Big( \arg\max_k S_k(x) = i \Big), \quad P_i = \frac{1}{B} \sum_{x \in \mathcal{B}} S_i(x).$$

The notation $\mathbb{1}(\cdot)$ denotes an indicator function that takes the value of one when its argument inside the parenthesis is true and zero otherwise. Note that $S_i(x)$ is the probability of assigning a particular sequence $x$ to expert $i$, while $P_i$ is the total probability fractions assigned to expert $i$ from all the sequences in the batch $\mathcal{B}$. Fedus et al. (2022) point out that the above load balancing loss could encourage uniform routing since it is minimized under a uniform distribution.

**Hyper-parameters** The hyper-parameters of learning $\text{KiC}_\text{Base}$ and $\text{KiC}_\text{Large}$ are listed in Table 9. In addition, we also list the hyper-parameters of single-task finetuning used in Table 10. Note that we set a maximum number of retrieved knowledge pieces to concatenate. If a knowledge-augmented input sequence exceeds the maximum input length, then it will be truncated.

| | Learning Rate | Max. Input Length | Max. Output Length | Batch Size | $\alpha$ | # epoch | Max. Knowledge Pieces |
|---|---|---|---|---|---|---|---|
| $\text{KiC}_\text{Base}$ | 5e−5 | 512 | 64 | 1024 | 0.05 | 5 | 10 |
| $\text{KiC}_\text{Large}$ | 5e−5 | 512 | 64 | 1024 | 0.01 | 5 | 10 |

Table 9: Hyper-parameters for $\text{KiC}_\text{Base}$ and $\text{KiC}_\text{Large}$.

| Model | Learning Rate | Max. Input Length | Max. Output Length | Batch Size | # epoch | Max. Knowledge Pieces |
|---|---|---|---|---|---|---|
| T5-LM-adapt$_\text{Base}$ | 2e−4 | 1024 | 512 | 16 | 10 | 5 |

Table 10: Hyper-parameters for single task fine-tuning.

**Computation cost** In Table 11, we provide the computation resources used for training $\text{T0}_\text{Base}$, $\text{T0}_\text{Large}$, $\text{KiC}_\text{Base}$ and $\text{KiC}_\text{Large}$ along with the total wall-clock time.

| | Hardware | Hours |
|---|---|---|
| $\text{T0}_\text{Base}$ | NVIDIA V100 $\times$ 64 | 21.2 |
| $\text{T0}_\text{Large}$ | NVIDIA V100 $\times$ 128 | 27.4 |
| $\text{KiC}_\text{Base}$ | NVIDIA V100 $\times$ 64 | 33.2 |
| $\text{KiC}_\text{Large}$ | NVIDIA V100 $\times$ 128 | 41.5 |

Table 11: Hardware and training time.

# B ADDITIONAL EXPERIMENTAL RESULTS

In this section, we provide additional experimental results and visualization results.

**Ablation studies of KiC** We now further examine the contribution of different components of the KiC model by performing extensive ablation studies. Specifically, we implement the following ablation models: (i) KiC without knowledge, (ii) KiC with an external memory that contains only plain text (English Wikipedia), (iii) KiC without knowledge-selector but retrieving from a mixture of

all knowledge categories, (iv) KiC with a task-adaptive selector, and (v) KiC without generalist. The results are reported in Table 12. First of all, it is important to leverage the knowledge-rich memory; when removing the knowledge memory or replacing it with a plain-text memory that consists of English Wikipedia, the performance would degrade greatly. Second, it is also important to use a knowledge selector to first pick a particular category of knowledge and then retrieve the relevant knowledge pieces from it. When we mix all the knowledge categories together with a single retriever, there would be a significant performance drop. The main reason is that different knowledge categories generally requires certain pre-filtering strategy during retrieval (see Appendix A.2). Furthermore, we also find that the instance-adaptive knowledge selector in our KiC model is crucial in achieving good performance. When we replace it with a task-adaptive selector, which picks a fixed knowledge category for all instances from the same task based on the task description, the performance is also noticeably worse. Finally, by comparing KiC without generalist to the original KiC, we also observe that there is a noticeable performance drop, which confirms the importance of allowing the model to ignore all external knowledge for some instances.

| Dataset | Task | $KiC_{Large}$ | | | w/o knowledge | | | /w plain texts | | | w/o selector | | | /w task-adaptive | | | w/o generalist | | |
|---|---|---|---|---|---|---|---|---|---|---|---|---|---|---|---|---|---|---|---|---|
| | | mean | med. | std | mean | med. | std | mean | med. | std | mean | med. | std | mean | med. | std | mean | med. | std |
| P3 | WSC | 62.6 | 65.4 | 8.3 | 57.6 | 59.1 | 5.9 | 53.3 | 52.9 | 7.7 | 63.5 | 64.4 | 1.8 | 62.2 | 63.9 | 5.0 | 62.8 | 64.4 | 7.5 |
| | Wino. XL | 54.1 | 55.3 | 2.4 | 50.4 | 50.5 | 0.3 | 52.2 | 52.2 | 0.4 | 52.5 | 52.2 | 1.0 | 53.8 | 54.7 | 2.2 | 54.2 | 54.9 | 2.0 |
| | $ANLI_{R1}$ | 36.7 | 36.3 | 1.8 | 31.3 | 30.5 | 2.0 | 34.0 | 33.9 | 0.8 | 31.6 | 31.5 | 1.1 | 33.1 | 32.2 | 2.3 | 35.2 | 34.6 | 1.8 |
| | $ANLI_{R2}$ | 34.9 | 35.0 | 1.4 | 32.8 | 32.7 | 0.6 | 33.7 | 33.9 | 0.8 | 32.8 | 32.8 | 0.8 | 33.7 | 33.1 | 1.8 | 35.1 | 34.9 | 1.1 |
| | $ANLI_{R3}$ | 37.6 | 37.6 | 2.5 | 34.0 | 33.8 | 0.8 | 37.0 | 37.8 | 1.7 | 33.9 | 34.0 | 1.2 | 35.5 | 35.6 | 1.6 | 37.0 | 37.1 | 1.5 |
| | CB | 57.5 | 67.9 | 22.9 | 52.0 | 60.7 | 23.0 | 59.3 | 69.6 | 21.6 | 52.7 | 62.5 | 20.3 | 54.0 | 60.7 | 22.1 | 56.8 | 66.1 | 21.1 |
| | RTE | 73.1 | 74.0 | 3.8 | 62.4 | 62.1 | 3.1 | 67.6 | 67.1 | 2.7 | 67.1 | 66.2 | 4.2 | 73.1 | 73.3 | 4.1 | 73.0 | 72.4 | 2.7 |
| | COPA | 81.7 | 85.3 | 6.8 | 71.6 | 73.5 | 8.4 | 71.1 | 73.5 | 7.1 | 75.7 | 79.0 | 7.3 | 77.6 | 82.0 | 6.9 | 83.9 | 85.2 | 6.2 |
| | HellaSwag | 29.7 | 29.6 | 0.9 | 25.7 | 25.7 | 0.4 | 26.9 | 26.7 | 0.9 | 28.4 | 28.4 | 0.4 | 29.5 | 29.3 | 1.1 | 28.6 | 28.3 | 0.9 |
| | StoryCloze | 93.9 | 94.4 | 1.2 | 84.5 | 84.1 | 1.8 | 86.9 | 86.7 | 1.2 | 90.4 | 91.1 | 1.6 | 89.0 | 90.0 | 2.0 | 93.6 | 94.3 | 1.4 |
| | WiC | 52.1 | 52.4 | 1.5 | 50.2 | 50.2 | 1.0 | 51.6 | 51.5 | 0.6 | 50.5 | 50.3 | 1.0 | 52.1 | 51.2 | 2.4 | 52.0 | 50.8 | 2.4 |
| | *Average* | 55.8 | 57.6 | | 50.2 | 51.2 | | 52.1 | 53.3 | | 52.6 | 53.9 | | 54.0 | 55.1 | | 55.6 | 56.6 | |
| MMLU | STEM | 30.7 | | | 28.2 | | | 29.1 | | | 28.7 | | | 29.2 | | | 30.3 | | |
| | Humanities | 38.3 | | | 31.9 | | | 32 | | | 36.3 | | | 35.3 | | | 37.3 | | |
| | Soc. Sci. | 43.6 | | | 33.2 | | | 36.6 | | | 42 | | | 40.5 | | | 42.5 | | |
| | Other | 44.8 | | | 33.8 | | | 36.4 | | | 42.7 | | | 41.8 | | | 43.8 | | |
| | *Average* | 39.4 | | | 31.8 | | | 33.5 | | | 37.4 | | | 36.7 | | | 38.5 | | |

Table 12: Ablation study of the $KiC_{Large}$ model. We consider the following four ablation models: (i) KiC without knowledge (i.e., T0), (ii) KiC with an external memory that contains only plain text (English Wikipedia), (iii) KiC without knowledge-selector but retrieving from a mixture of all knowledge categories, (iv) KiC with a task-adaptive selector, and (v) KiC without generalist. We report the mean, median and standard deviation for P3 tasks over different templates. For MMLU, we report the results on the test set, just like other works in the literature.

**Which categories of knowledge are useful for an unseen task?** To understand what kind of knowledge categories are retrieved to help a particular task, we report the distribution of the selected knowledge by $KiC_{Large}$ for each task in Figure 4. The results show that most of the knowledge categories are useful for different tasks. And the knowledge selector is able to pick the most helpful knowledge type for solving its current task. For example, in Word-in-Context (WiC) task, the model mostly retrieves from the dictionary knowledge to help it disambiguate different word senses. In StoryCloze task, it relies more heavily on commonsense knowledge to complete the story ending. For MMLU tasks, since they cover a large variety of subjects (i.e., 57 subjects), it is not surprising that it needs more diverse categories of knowledge. In addition, the results further show that the generalist in KiC is also very important as the model would frequently choose it when solving different tasks. It demonstrates the necessity of allowing the model to ignore all knowledge categories for some instances. Finally, we would like to highlight that we never use any direct supervision to train the knowledge selector. Instead, it learns to make such decisions from the distant supervision of predicting the correct answer. This is valuable because learning to identify the most helpful knowledge for solving a particular task is an important step toward general intelligence. More importantly, the results also confirm the effectiveness of our learning strategy based on our KiC-MoE equivalence.

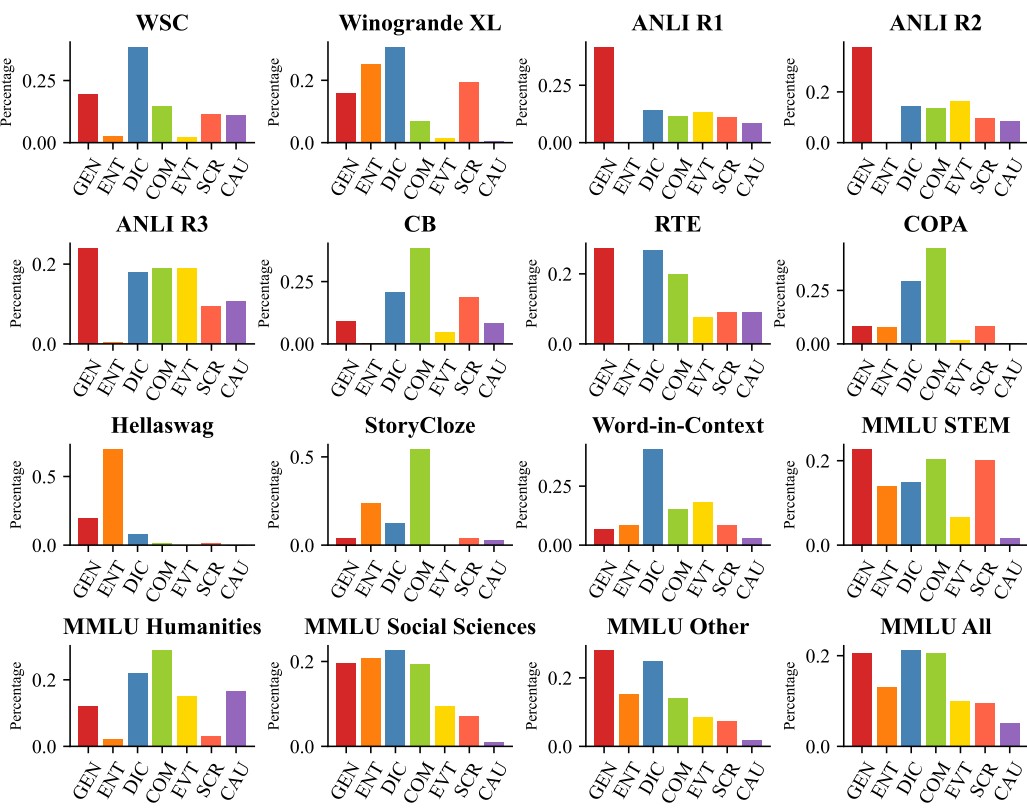

Figure 4: The distribution of the selected knowledge categories for each task. We examine the following categories of knowledge: entity (ENT), dictionary (DIC), commonsense (COM), event (EVT), script (SCR), or causality (CAU) knowledge. In addition, the generalist (GEN) means that we do not choose any external knowledge but make predictions based solely on the input query.

**Full results for zero-shot performance** In Table 13, we provide the full zero-shot results on holdout unseen tasks, where we report both mean and median results together. The reason that we report both the mean and median is to be consistent with the results in the T0 paper (Sanh et al., 2022), where they report both metrics. In the main paper, we only keep the median results for brevity.

| Models | Params | Coreference | | NLI | | | | | Completion | | | WSD |
| | | WSC | Wino. XL | ANLI$_{R1}$ | ANLI$_{R2}$ | ANLI$_{R3}$ | CB | RTE | COPA | HellaSwag | StoryCloze | WiC |
|---|---|---|---|---|---|---|---|---|---|---|---|---|
| BERT | 0.34B | 53.2/53.4 | 49.7/49.5 | 34.1/33.8 | 33.8/33.8 | 33.2/33.2 | 45.0/48.2 | 48.3/48.4 | 48.9/49.0 | 25.5/25.5 | 50.1/50.1 | 50.3/50.2 |
| RoBERTa | 0.35B | 38.5/37.0 | 49.1/48.9 | 33.6/33.4 | 33.6/33.4 | 33.5/33.3 | 44.0/42.9 | 53.8/54.1 | 56.8/56.0 | 22.9/22.4 | 48.5/48.5 | 50.4/50.0 |
| GPT-Neo | 2.7B | 49.2/45.2 | 50.4/50.8 | 34.0/33.5 | 33.5/33.3 | 33.5/33.4 | 36.4/48.2 | 51.1/51.1 | 50.5/50.5 | 24.8/25.0 | 54.0/54.6 | 52.2/52.5 |
| GPT-J | 6B | 44.6/40.4 | 48.9/48.5 | 33.7/34.0 | 34.0/33.7 | 33.7/33.6 | 26.6/28.6 | 49.6/50.5 | 54.9/56.0 | 24.7/24.7 | 53.1/53.3 | 52.4/51.0 |
| OPT | 30B | 53.4/63.5 | 48.5/48.4 | 33.2/33.3 | 33.3/33.3 | 33.4/33.4 | 40.6/50.0 | 47.8/47.3 | 52.5/52.0 | 24.5/24.4 | 55.5/55.3 | 50.0/50.0 |
| GPT-NeoX | 20B | 56.0/60.6 | 49.1/48.9 | 33.5/33.4 | 33.6/33.4 | 33.6/33.6 | 30.6/30.4 | 49.3/48.4 | 45.9/44.5 | 24.9/25.0 | 53.1/53.5 | 50.8/49.9 |
| T0$_{Base}$ | 0.22B | 58.6/61.1 | 50.7/50.6 | 31.7/32.2 | 33.0/33.0 | 34.1/34.2 | 44.3/53.6 | 64.4/64.1 | 65.8/65.7 | 25.7/25.7 | 80.6/80.6 | 50.6/50.7 |
| T0$_{Large}$ | 0.77B | 57.6/59.1 | 50.4/50.5 | 31.3/30.5 | 32.8/32.7 | 34.0/33.8 | 52.0/60.7 | 62.4/62.1 | 71.6/73.5 | 25.7/25.7 | 84.5/84.1 | 50.2/50.2 |
| T0$_{XL}$ | 3B | 65.1/64.4 | 51.0/50.5 | 33.8/33.7 | 33.1/33.4 | 33.3/33.3 | 45.4/50.0 | 64.6/64.1 | 72.4/74.9 | 27.3/27.5 | 84.0/85.1 | 50.7/50.4 |
| T0$_{XXL}$ | 11B | 61.5/64.4 | 59.9/60.5 | 43.6/44.7 | 38.7/39.4 | 41.3/42.4 | 70.1/78.6 | 80.8/81.2 | 90.0/90.8 | 33.6/33.7 | 92.4/94.7 | 56.6/57.2 |
| KiC$_{Small}$ | 0.6B | 61.3/63.5 | 51.2/51.1 | 33.5/33.3 | 33.6/33.3 | 33.6/33.6 | 40.4/44.6 | 48.1/47.3 | 48.4/48.0 | 25.3/25.4 | 58.1/57.7 | 50.1/50.0 |
| KiC$_{Base}$ | 0.22B | 63.7/63.5 | 49.9/50.0 | 29.3/28.4 | 31.5/30.9 | 33.0/32.8 | 52.9/58.9 | 66.9/66.8 | 61.9/65.0 | 26.2/26.1 | 82.4/82.6 | 51.1/50.2 |
| KiC$_{Large}$ | 0.77B | 62.6/65.4 | 54.1/55.3 | 36.7/36.3 | 34.9/35.0 | 37.6/37.6 | 57.5/67.9 | 73.1/74.0 | 81.7/85.3 | 29.7/29.6 | 93.9/94.4 | 52.1/52.4 |

Table 13: Full zero-shot evaluation results on holdout unseen tasks. We report mean/median accuracy (%) over all prompts for each task.

# C  DESCRIPTIONS OF 35 EVALUATION TASKS IN TABLE 1

We show the description of all evaluation tasks in Table 14. We categorize these tasks in the same way as the T0 paper (Sanh et al., 2022), with a brief explanation for each category of tasks. For more detailed information, please refer to the original papers listed in Table 14.

| Category | Tasks | Task Description |
|---|---|---|
| Coreference | WSC (Levesque et al., 2012), Winogrande (debiased and XL) (Sakaguchi et al., 2021) | Each instance in the pronoun coreference task has a target pronoun and two candidates. The task requires models to link the target pronoun to the correct mention by conducting commonsense reasoning. |
| NLI | CB (De Marneffe et al., 2019) and RTE (Wang et al., 2019) | Natural language inference is the task of determining whether a "hypothesis" is true (entailment), false (contradiction), or undetermined (neutral) given a "premise." |
| Paraphrase | MRPC (Dolan & Brockett, 2005), QQP (Wang et al., 2018), PAWS (Zhang et al., 2019a) | Paraphrase identification (PI) is concerned with the ability to identify alternative linguistic expressions of the same meaning at different textual levels. |
| Closed QA | ARC (Easy and Challenge) (Clark et al., 2018), WikiQA (Yang et al., 2015) | In the closed book QA, each question is associated with a document, and the models are required to answer the question with the document. |
| Extractive QA | ReCoRD (Zhang et al., 2018) | Extractive QA aims to extract a text span from the passage to answer the questions. |
| Multiple Choice QA | CoS-E v1.11 (Rajani et al., 2019), CosmosQA (Huang et al., 2019), DREAM (Sun et al., 2019), OpenBookQA (Mihaylov et al., 2018), PIQA (Bisk et al., 2020), QASC (Khot et al., 2020), QuAIL (Rogers et al., 2020), QuaRTz (Tafjord et al., 2019), RACE (Middle and Hign) (Lai et al., 2017), SciQ (Welbl et al., 2017), SocialIQA (Sap et al., 2019), BoolQ (Clark et al., 2019), MultiRC (Khashabi et al., 2018), WikiHop (Welbl et al., 2018), WIQA (Tandon et al., 2019) | In multiple choice QA, each question is associated with several answers, and the models are required to select the correct one/ones. |
| Sentiment Analysis | IMDB (Maas et al., 2011) and Rotten Tomatoes (Pang & Lee, 2005) | Sentiment analysis aims at predicting the sentiment attitude of a text span (mostly sentences or reviews). |
| Sentence Completion | HellaSwag (Zellers et al., 2019), COPA (Roemmele et al., 2011), Story Cloze(Mostafazadeh et al., 2016) | Decide which sentence is the most plausible ending of the given sentence(s). |
| Topic Classification | AG News (Del Corso et al., 2005) and DBpedia14 (Lehmann et al., 2015). | Classify a given sentence into one of the predefined topic categories. |
| WSD | WiC (Pilehvar et al., 2019) | The WSD task provides two sentences containing the same lemma word and asks whether the two target words have the same meaning. |

Table 14: Task descriptions of all selected tasks.

# D    CASE STUDY OF RETRIEVED KNOWLEDGE

We show examples of retrieved knowledge in Table 15. Different knowledge plays critical roles in various tasks. For instance, in the Hellaswag task, the model can predict that a person will mow the lawn because it finds the commonsense knowledge that a "lawn mover" is used for cutting grass. Similarly, in the WiC task, the model knows that the two "pockets" are different with the help of a detailed explanation of different synsets of the word "pocket." Last but not least, in the Winogrande task, the model can successfully know that burglary is more likely to be investigated because it finds the event knowledge that burglary is often concluded by an investigator.

| | |
|---|---|
| **Task** | Hellaswag |
| **Input** | A first person view is seen of a man riding a riding lawn mower. he...How does the description likely end?Ending 1: takes turns quickly, mowing the lawn.Ending 2: creates a large puddle of water and a high rush of water around him as he heads back and forth back and forth.Ending 3: moves all around while there is a crowd watching.Ending 4: talks about how to properly ride an object while another man climbs up on the back of him. |
| **Output** | Ending 1 |
| **Knowledge Type** | Commonsense |
| **Knowledge Piece** | *lawn mower UsedFor cutting grass; ride on RelatedTo lawn mower* |
| **Task** | WiC |
| **Input** | Sentence 1: Lydia put the change in her left pocket.Sentence 2: Lydia pocketed the change.Determine whether the word "pocket" is used in the same sense in both sentences. Yes or no? |
| **Output** | no |
| **Knowledge Type** | Lexicon |
| **Knowledge Piece** | *pocket: A bag stitched to an item of clothing, used for carrying small items. Such a receptacle seen as housing someone's money; hence, financial resources.* |
| **Task** | Winogrande XL |
| **Input** | She decided to report the accident and the burglary, but the _ required much more investigation. In the previous sentence, does _ refer to burglary or accident? |
| **Output** | burglary |
| **Knowledge Type** | Event |
| **Knowledge Piece** | *the investigator conclude Co_Occurrence it have be a burglary* |

Table 15: Examples of the improved instances and the corresponding selected knowledge.

# E PROMPT TEMPLATES FOR KNOWLEDGE-IN-CONTEXT

We provide the prompt templates for training and evaluating our KiC system. Note that we use the same naming convention for the templates as the original P3 dataset (Sanh et al., 2022).

| Dataset | Template |
|---|---|
| adversarial_qa/dbert | `answer_the_following_q`
`based_on`
`generate_question`
`question_context_answer`
`tell_what_it_is` |
| adversarial_qa/dbidaf | `answer_the_following_q`
`based_on`
`generate_question`
`question_context_answer`
`tell_what_it_is` |
| adversarial_qa/droberta | `answer_the_following_q`
`based_on`
`generate_question`
`question_context_answer`
`tell_what_it_is` |
| cos_e_v1.11 | `aligned_with_common_sense`
`description_question_option_id`
`description_question_option_text`
`explain_why_human`
`generate_explanation_given_text`
`i_think`
`question_description_option_id`
`question_description_option_text`
`question_option_description_id`
`question_option_description_text`
`rationale` |
| cosmos_qa | `context_answer_to_question`
`context_description_question_answer_id`
`context_description_question_answer_text`
`context_description_question_text`
`context_question_description_answer_id`
`context_question_description_answer_text`
`context_question_description_text`
`description_context_question_answer_id`
`description_context_question_answer_text`
`description_context_question_text`
`no_prompt_id`
`no_prompt_text`
`only_question_answer` |
| dream | `answer-to-dialogue`
`baseline`
`generate-first-utterance`
`generate-last-utterance`
`read_the_following_conversation_and_answer_`
`the_question` |
| glue_mrpc | `equivalent`
`generate_paraphrase`
`generate_sentence`
`paraphrase`
`replace`
`same_thing`
`want_to_know` |
| glue_qqp | `answer`
`duplicate`
`duplicate_or_not`
`meaning`
`quora` |

| | |
|---|---|
| | `same_thing` |
| imdb | `Movie_Expressed_Sentiment`
`Movie_Expressed_Sentiment_2`
`Negation_template_for_positive_and_negative`
`Reviewer_Enjoyment`
`Reviewer_Enjoyment_Yes_No`
`Reviewer_Expressed_Sentiment`
`Reviewer_Opinion_bad_good_choices`
`Reviewer_Sentiment_Feeling`
`Sentiment_with_choices_`
`Text_Expressed_Sentiment`
`Writer_Expressed_Sentiment` |
| paws_labeled_final | `Concatenation`
`Concatenation-no-label`
`Meaning`
`Meaning-no-label`
`PAWS-ANLI_GPT3`
`PAWS-ANLI_GPT3-no-label`
`Rewrite`
`Rewrite-no-label`
`context-question`
`context-question-no-label`
`paraphrase-task`
`task_description-no-label` |
| qasc | `is_correct_1`
`is_correct_2`
`qa_with_combined_facts_1`
`qa_with_separated_facts_1`
`qa_with_separated_facts_2`
`qa_with_separated_facts_3`
`qa_with_separated_facts_4`
`qa_with_separated_facts_5` |
| quail | `context_description_question_answer_id`
`context_description_question_answer_text`
`context_description_question_text`
`context_question_answer_description_id`
`context_question_answer_description_text`
`context_question_description_answer_id`
`context_question_description_answer_text`
`context_question_description_text`
`description_context_question_answer_id`
`description_context_question_answer_text`
`description_context_question_text`
`no_prompt_id`
`no_prompt_text` |
| quarel | `choose_between`
`do_not_use`
`heres_a_story`
`logic_test`
`testing_students` |
| quartz | `answer_question_based_on`
`answer_question_below`
`given_the_fact_answer_the_q`
`having_read_above_passage`
`paragraph_question_plain_concat`
`read_passage_below_choose`
`use_info_from_paragraph_question`
`use_info_from_question_paragraph` |
| quoref | `Answer_Friend_Question`
`Answer_Question_Given_Context`
`Answer_Test`
`Context_Contains_Answer`
`Find_Answer`
`Found_Context_Online` |

|  | |
|---|---|
|  | Given_Context_Answer_Question
Guess_Answer
Guess_Title_For_Context
Read_And_Extract_
What_Is_The_Answer |
| ropes | background_new_situation_answer
background_situation_middle
given_background_situation
new_situation_background_answer
plain_background_situation
plain_bottom_hint
plain_no_background
prompt_beginning
prompt_bottom_hint_beginning
prompt_bottom_no_hint
prompt_mix
read_background_situation |
| rotten_tomatoes | Movie_Expressed_Sentiment
Movie_Expressed_Sentiment_2
Reviewer_Enjoyment
Reviewer_Enjoyment_Yes_No
Reviewer_Expressed_Sentiment
Reviewer_Opinion_bad_good_choices
Reviewer_Sentiment_Feeling
Sentiment_with_choices_
Text_Expressed_Sentiment
Writer_Expressed_Sentiment |
| samsum | Generate_a_summary_for_this_dialogue
Given_the_above_dialogue_write_a_summary
Sum_up_the_following_dialogue
Summarize:
Summarize_this_dialogue:
To_sum_up_this_dialog
Write_a_dialogue_that_match_this_summary |
| sciq | Direct_Question
Direct_Question_(Closed_Book)
Multiple_Choice
Multiple_Choice_(Closed_Book)
Multiple_Choice_Question_First |
| social_i_qa | Check_if_a_random_answer_is_valid_or_not
Generate_answer
Generate_the_question_from_the_answer
I_was_wondering
Show_choices_and_generate_answer
Show_choices_and_generate_index |
| trec | fine_grained_ABBR
fine_grained_ABBR_context_first
fine_grained_DESC
fine_grained_DESC_context_first
fine_grained_ENTY
fine_grained_HUM
fine_grained_HUM_context_first
fine_grained_LOC
fine_grained_LOC_context_first
fine_grained_NUM
fine_grained_NUM_context_first
fine_grained_open
fine_grained_open_context_first
pick_the_best_descriptor
trec1
trec2
what_category_best_describe
which_category_best_describes |
| wiki_hop_original | choose_best_object_affirmative_1 |

| | |
|---|---|
| | ```choose_best_object_affirmative_2```
```choose_best_object_affirmative_3```
```choose_best_object_interrogative_1```
```choose_best_object_interrogative_2```
```explain_relation```
```generate_object```
```generate_subject```
```generate_subject_and_object``` |
| wiki_qa | ```Decide_good_answer```
```Direct_Answer_to_Question```
```Generate_Question_from_Topic```
```Is_This_True?```
```Jeopardy_style```
```Topic_Prediction_-_Answer_Only```
```Topic_Prediction_-_Question_Only```
```Topic_Prediction_-_Question_and_Answer_Pair```
```automatic_system```
```exercise```
```found_on_google``` |
| wiqa | ```does_the_supposed_perturbation_have_an_effect```
```effect_with_label_answer```
```effect_with_string_answer```
```what_is_the_final_step_of_the_following_process```
```what_is_the_missing_first_step```
```what_might_be_the_first_step_of_the_process```
```what_might_be_the_last_step_of_the_process```
```which_of_the_following_is_the_supposed_perturbation``` |

Table 16: All used training datasets and templates from P3 (Sanh et al., 2022) for KiC.

| Dataset | Template |
|---|---|
| anli | ```GPT-3_style```
```MNLI_crowdsource```
```always_sometimes_never```
```based_on_the_previous_passage```
```can_we_infer```
```claim_true_false_inconclusive```
```consider_always_sometimes_never```
```does_it_follow_that```
```does_this_imply```
```guaranteed_possible_impossible```
```guaranteed_true```
```justified_in_saying```
```must_be_true```
```should_assume```
```take_the_following_as_truth``` |
| hellaswag | ```Predict_ending_with_hint```
```Randomized_prompts_template```
```complete_first_then```
```if_begins_how_continues``` |
| super_glue_cb | ```GPT-3_style```
```MNLI_crowdsource```
```always_sometimes_never```
```based_on_the_previous_passage```
```can_we_infer```
```claim_true_false_inconclusive```
```consider_always_sometimes_never```
```does_it_follow_that```
```does_this_imply```
```guaranteed_possible_impossible```
```guaranteed_true```
```justified_in_saying``` |

| | |
|---|---|
| | must_be_true
should_assume
take_the_following_as_truth |
| super_glue_copa | C1_or_C2?_premise,_so_because...
best_option
cause_effect
choose
exercise
i_am_hesitating
more_likely
plausible_alternatives
...As_a_result,_C1_or_C2?
...What_could_happen_next,_C1_or_C2?
...which_may_be_caused_by
...why?_C1_or_C2 |
| super_glue_rte | GPT-3_style
MNLI_crowdsource
based_on_the_previous_passage
can_we_infer
does_it_follow_that
does_this_imply
guaranteed_true
justified_in_saying
must_be_true
should_assume |
| super_glue_wic | GPT-3-prompt
GPT-3-prompt-with-label
affirmation_true_or_false
grammar_homework
polysemous
question-context
question-context-meaning
question-context-meaning-with-label
same_sense
similar-sense |
| super_glue_wsc.fixed | GPT-3_Style
I_think_they_mean
Who_or_what_is_are
by_p_they_mean
does_p_stand_for
does_the_pronoun_refer_to
in_other_words
p_is_are_r
replaced_with
the_pronoun_refers_to |
| story_cloze_2016 | Answer_Given_options
Choose_Story_Ending
Movie_What_Happens_Next
Novel_Correct_Ending
Story_Continuation_and_Options |
| winogrande_winogrande_xl | Replace
does_underscore_refer_to
fill_in_the_blank
stand_for
underscore_refer_to |
| mmlu_all | heres_a_problem
i_am_hesitating
multiple_choice
pick_false_options
pick_the_most_correct_option
qa_options |
| mmlu_humanities | heres_a_problem
i_am_hesitating
multiple_choice |

| | |
|---|---|
| | `pick_false_options` 
 `pick_the_most_correct_option` 
 `qa_options` |
| mmlu_other | `heres_a_problem` 
 `i_am_hesitating` 
 `multiple_choice` 
 `pick_false_options` 
 `pick_the_most_correct_option` 
 `qa_options` |
| mmlu_social_sciences | `heres_a_problem` 
 `i_am_hesitating` 
 `multiple_choice` 
 `pick_false_options` 
 `pick_the_most_correct_option` 
 `qa_options` |
| mmlu_stem | `heres_a_problem` 
 `i_am_hesitating` 
 `multiple_choice` 
 `pick_false_options` 
 `pick_the_most_correct_option` 
 `qa_options` |

Table 17: All used evaluation datasets and templates from P3 (Sanh et al., 2022) and MMLU (Hendrycks et al., 2020) for KiC. Note that the original MMLU tasks do not include templates, we use the templates of ai2_arc/ARC_Challenge in P3 for MMLU evaluation.

