# OpenReview forum: "Knowledge-in-Context: Towards Knowledgeable Semi-Parametric Language Models"
_ICLR.cc/2023/Conference — ICLR 2023 notable top 25%_

### Official Review · Reviewer_mSmi · 2022-10-24

**Confidence:** 4
**Correctness:** 3
**Technical Novelty And Significance:** 3
**Empirical Novelty And Significance:** 3
**Recommendation:** 6

**Clarity, Quality, Novelty And Reproducibility:**

This paper is clear about the contribution and the proposed method.
Although there are many lines of work related to semi-parametric zero-shot task adaptation, the method proposed in this paper is novel enough.

**Strength And Weaknesses:**

Strength:
1. The paper is well-written and clear about the contributions.
2. The performance of KiC shows significant improvement compared to baselines.
3. The experiments are extensive, containing zero-shot task transfer (MMLU and datasets evaluated in T0), and in-domain evaluation.

Weakness:
This paper lacks detailed ablation and analysis of the proposed method.
1. The performance of retrieving from a mixture of 6 categories (treating a mixture of 6 categories as a single large corpus) is needed to show the effectiveness of MoE architecture for knowledge selection. A naive mixture of 6 categories would not need any knowledge selection step.
2. Please also report the standard deviation across different evaluation prompts for each task in Table 2. Prompt sensitivity is also important in addition to median accuracy.
3. For each task, what is the occurrence of selection of each knowledge category for Table 2? This might answer the question of "why does KiC works?", if there are some patterns of the target task and selected expert.
4. Is there any ablation result showing the necessity of generalist (no external memory)? How does the performance vary if the generalist is absent?

Questions:
1. Are there any results of KiC on a 3B (or larger) scale?
2. Is there any quantitative result of showing the benefit of using structured knowledge resources than plain text such as Wikipedia or C4 dataset? If the effect is similar, using plain text would be simpler because it does not require any knowledge selector.

**Summary Of The Paper:**

This paper introduces Knowledge-in-Context (KiC), a semi-parametric LM consisting of a parametric text-to-text LM with a knowledge-rich external memory and a knowledge selector. During inference, the knowledge selector determines the sequence-to-expert assignment and a retriever retrieves the most relevant sequence from the expert. The retrieved sequence is concatenated to the data instance and goes through the text-to-text LM. Results show that KiC outperforms LMs that are 4-39x larger by a significant margin.

**Summary Of The Review:**

This paper proposes KiC, an effective semi-parametric LM that outperforms much larger parametric LMs on various downstream tasks. The paper is well-written and the proposed method is novel enough. However, this paper lacks detailed analysis and ablation. Ablation studies on how each component of the proposed method benefits the target downstream task are needed, especially.

---

> ### Author Response · Authors · 2022-11-18
> **Response to Reviewer mSmi (1/2)**
>
> **Q1: Treating a mixture of 6 categories as a single large corpus.**
>
> Different knowledge categories generally have very different characteristics, and may need knowledge-specific processing in order to accurately retrieve them for each instance. For example, to retrieve the entity knowledge, we need to apply entity linking to the query and pre-filter the candidate knowledge pieces before using MIPS search (See our newly added Appendix A.2 for more details). Besides, retrieving from a mixed corpus may generate vastly different scores for candidate knowledge pieces from different knowledge categories. For example, ranking scores of script knowledge are generally lower than other knowledge categories due to the relatively long length. Therefore, such knowledge pieces are very unlikely to be selected in a mixed corpus. For this reason, it is more natural to have a knowledge selector to pick the most relevant knowledge and then retrieve the most relevant knowledge pieces from it. To demonstrate the importance of the knowledge selector, we have added a new ablation baseline that mixes all the knowledge categories together and use MIPS to retrieve the relevant knowledge pieces from them --- see results in Appendix B. The result shows that such a strategy would lead to a significant performance drop, which confirms the importance of using our knowledge selector. Ideally, integrating all different kinds of knowledge together would require a universally powerful knowledge representation and retrieval paradigm, which is still an open research problem. Therefore, we leave this as future work.
>
> **Q2: Report the standard deviation across different evaluation prompts for each task.**
>
> Thanks for the suggestion. In our updated paper, we have added standard deviation in Table 2 and Table 4. For MMLU (Table 3), following standard approaches, we only choose the prompt that yields the best accuracy on the validation set.
>
> **Q3: What is the occurrence of selection of each knowledge category?**
>
> Thanks for the suggestion. In Appendix B of our updated paper, we have added a number of plots that illustrate the distributions of the selected knowledge for each task (Figure 4). The results show that most of the knowledge categories are useful for different tasks. And the knowledge selector is able to pick the most helpful knowledge type for solving its current task. For example, in Word-in-Context (WiC) task, the model mostly retrieves from the dictionary knowledge to help it disambiguate different word senses. In StoryCloze task, it relies more heavily on commonsense knowledge to complete the story ending. For MMLU tasks, since they cover a large variety of subjects (i.e., 57 subjects), it is not surprising that it needs more diverse categories of knowledge. In addition, the results further show that the generalist in KiC is also very important as the model would frequently choose it when solving different tasks. It demonstrates the necessity of allowing the model to ignore all external knowledge categories for some instances.

---

> > ### Author Response · Authors · 2022-11-20
> > **Response to Reviewer mSmi (2/2)**
> >
> > **Q4: Ablation result of showing the necessity of generalist. Result of showing the benefit of using structured knowledge resources than plain text such as Wikipedia or C4 dataset.**
> >
> > Thanks for the suggestion. In our updated paper, we have added a new Appendix B to include your suggested ablation studies along with others. Specifically, we have added the following baselines: (i) KiC without knowledge, (ii) KiC with an external memory that contains only plain text (English Wikipedia), (iii) KiC without knowledge-selector but retrieving from a mixture of all knowledge categories, (iv) KiC with a task-adaptive selector, and (v) KiC without generalist. The results are reported in Table 12. First of all, it is important to leverage the knowledge-rich memory; when removing the knowledge memory or replacing it with a plain-text memory that consists of English Wikipedia, the performance would degrade greatly. Second, it is also important to use a knowledge selector to first pick a particular category of knowledge and then retrieve the relevant knowledge pieces from it. When we mix all the knowledge categories together with a single retriever, there would be a significant performance drop. The main reason is that different knowledge categories generally require certain pre-filtering strategy during retrieval (see Appendix A.2). Furthermore, we also find that the instance-adaptive knowledge selector in our KiC model is crucial in achieving good performance. When we replace it with a task-adaptive selector, which picks a fixed knowledge category for all instances from the same task based on the task description, the performance is also noticeably worse. Finally, by comparing **KiC without generalist** to the original KiC, we also observe that there is a noticeable performance drop, which confirms the importance of allowing the model to ignore all external knowledge for some instances.
> >
> > **Q5: Are there any results of KiC on a 3B (or larger) scale?**
> >
> > We currently do not have the KiC-3B model ready due to limited computing power. We plan to extend to the KiC-3B and larger models in future work.

---

> > > ### Comment · Reviewer_mSmi · 2022-11-22
> > > **Response to Authors**
> > >
> > > Thank you for your reply and additional experiments!
> > >
> > > I still have one concern regarding the standard deviation results in Table 2 and Table 4.
> > > The standard deviation seems to be quite large for KiC (large) compared to other baselines.
> > > Are there any comments on this?
> > > For the worst-case prompt (prompt that leads to the worst performance), does the performance underperform other baselines' (especially T0) worst-case prompt performance?

---

> > > > ### Author Response · Authors · 2022-11-24
> > > > **Response to Reviewer mSmi**
> > > >
> > > > Thank you for your follow-up question. First, we would like to highlight that KiC-large improves the standard deviation over some tasks (e.g., COPA and StoryCloze). On the other hand, we found that the large standard deviation of KiC-large model in a few tasks (e.g., WSC and Winogrande) is mainly caused by a few prompt templates that always perform very badly compared to others. For example, in the following table, we report the full results of WSC for each used prompt template and compare the performance between KiC_large and T0_large.
> > > >
> > > >
> > > > | Template|	KiC_large|	T0_large|	$\Delta$ |
> > > > |--|--|--|--|
> > > > |p_is_are_r|	62.50%|	60.58%|	`1.92%`|
> > > > |does_p_stand_for|	69.23%|	59.62%|	`9.62%`|
> > > > |the_pronoun_refers_to|	66.35%|	54.81%|	`11.54%`|
> > > > |in_other_words|	61.54%|	57.69%|	`3.85%`|
> > > > |GPT-3_Style|	38.46%|	48.08%|	`-9.62%`|
> > > > |does_the_pronoun_refer_to|	64.42%|	64.42%|	`0.00%`|
> > > > |I_think_they_mean|	65.38%|	61.54%|	`3.85%`|
> > > > |by_p_they_mean|	66.35%|	58.65%|	`7.69%`|
> > > > |Who_or_what_is_are|	66.35%|	46.15%|	`20.19%`|
> > > > |replaced_with|	65.38%|	64.42%|	`0.96%`|
> > > > |*mean*	|62.60%|	57.60%|	`5.00%`|
> > > > |*median*	|65.38%|	59.13%|	`6.25%`|
> > > >
> > > > KiC can improve substantially over some poorly performing templates (e.g., "Who_or_what_is_are" and "the_pronoun_refers_to"). However, it does not consistently improve some extremely poor templates (e.g., "GPT-3_Style").
> > > > The following shows an example of the worst-case template "GPT-3_Style" (the highlighted part is the template and the italic part is the original input):
> > > >
> > > > > **Passage:** *Bernard , who had not told the government official that he was less than 21 when he filed for a homestead claim, did not consider that he had done anything dishonest. Still, anyone who knew that he was 19 years old could take his claim away from him .*
> > > > >
> > > > > **Question: In the passage above, does the pronoun** *"him"* **refer to** *anyone* **?**
> > > > >
> > > > > **Answer:**
> > > >
> > > > We hypothesize that the reason for the poor performance may be the misleading effect caused by the keywords "passage", "question" and "answer"; that is, they mislead the model into thinking that this task is a multiple-choice reading comprehension question rather than a binary classification question.
> > > >
> > > >
> > > > Such behavior has also been observed in the T0 paper as well. The following table shows the results (median (%) and standard deviation (%)) for T0 model variants on unseen tasks. Increasing model size (T0_11B vs T0_3B) and adding more training data (T0+_11B vs T0_11B) generally can improve the overall (median) performance. However, the spread (standard deviation) is not consistently improved. As the authors pointed out in the T0 paper, the main reason is that some prompts always perform badly. Therefore, the spread is stretched larger when other prompts improve.
> > > >
> > > > | | WSC |Winogrande (XL)  | ANLI R1 | ANLI R2 | ANLI R3 | CB |RTE  | COPA | HellaSwag | StoryCloze | Wic |
> > > > |--|--|--|--|--|--|--|--|--|--|--|--|
> > > > |T0_3B |64.4	`2.7`|50.5	`1.2`|33.6	`0.9`|33.4	`1.2`|33.3	`0.4`|50.0	`15.9`|64.1	`3.5`|74.9	`8.7`|27.5	`1.0`|85.1	`3.2`|50.4	`0.9`|
> > > > |T0_11B| 64.4	`6.3`|60.5	`2.5`|44.7	`3.6`|39.4	`2.2`|42.4	`3.0`|78.6	`18.5`|81.2	`3.7`|90.8	`4.1`|33.7	`0.5`|94.7	`4.7`|57.2	`1.8`|
> > > > |T0+_11B| 64.4	`5.4`|61.7	`1.8`|45.8	`5.2`|41.1	`3.2`|41.2	`3.8`|71.4	`25.9`|65.0	`5.9`|93.9	`3.1`|- |- |55.5	`3.7`|
> > > >
> > > > Overall, we believe that improving the prompt robustness on unseen tasks is still an open research question and we leave it to future work.

---

### Official Review · Reviewer_TPmn · 2022-10-25

**Confidence:** 5
**Correctness:** 4
**Technical Novelty And Significance:** 4
**Empirical Novelty And Significance:** 4
**Recommendation:** 8

**Clarity, Quality, Novelty And Reproducibility:**

The paper is very easy to read will clear explantations in the technical parts such as the explanations regarding the MoE layer. I think the proposed method is very novel in the sense that they utilized a MoE layer to retrieve from multiple external sources and that they utilized this to tackle the important task of generalizing to unseen tasks.

**Strength And Weaknesses:**

The strength of the paper is that it utilizes a novel semi-parametric language model architecture that retrieves from multiple knowledge sources dynamically via an MoE layer. This simple yet effective approach boosts zero-shot task generalization results significantly. Dividing the knowledge sources into 6 different resources helps in different aspects of solving NLP tasks, which can be decided at a instance level.

The weakness is that it does not show any computational comparison compared to prior multitask prompted finetuning approaches. Initial thought is that the proposed method might require much more computation (fine-tuning stage) since the MoE layer has to be trained to be able to dynamically select which knowledge source to route to AND also train the underlying LM.


**Summary Of The Paper:**

This paper tackles the problem of zero-shot task generalization to unseen tasks using a semi-parametric approach. In order to achieve this, they construct 6 different knowledge-rich external memory consisting of Dictionary, Commonsense, Entity, Event, Script, and Causality. Like previous work, they perform multitask prompted fine-tuning on 40+ NLP tasks but while retrieving from external knowledge sources to perform the task. Since there are six knowledge sources, they train a MoE layer that dynamically routes to which source to retrieve the external knowledge from. This approach coined Knowledge-in-Context (KiC) enables a 770M LM to easily outperform LMs that are 4-39x larger by a significant margin.

**Summary Of The Review:**

This paper suggests a novel semiparametric architecture, retrieving from multiple fine-grained to coarse-grained knowledge sources to solve unseen tasks and achieving significant performance enhancement compared to previous approaches while having a much smaller number of parameters. Thus, I highly recommend this paper be accepted at this conference.

---

> ### Author Response · Authors · 2022-11-18
> **Response to Reviewer TPmn**
>
> **Q1: Computational comparison to prior multitask prompted finetuning approaches.**
>
> Thanks for the suggestion. In our updated paper, we have added training time for T0-base and T0-large in Table 11 of Appendix. We would like to highlight that our MoE layer is not very heavy: it consists of an encoder that is shared with the backbone T5 model followed by a mean-pooling layer and a $(K+1)$-class linear classifier --- see revised Section 2.3 for our construction of knowledge selector. Therefore, the training time of KiC-large is roughly (our distributed training platform has communication cost) 51% longer than T0-large (without knowledge and knowledge selector). The overhead is mainly caused by the shared encoder, which is applied one extra time for the selector.

---

### Official Review · Reviewer_6HCz · 2022-10-25

**Confidence:** 4
**Correctness:** 3
**Technical Novelty And Significance:** 3
**Empirical Novelty And Significance:** 3
**Recommendation:** 6

**Clarity, Quality, Novelty And Reproducibility:**

Several important details of the method are missing, making it less clear and the experimental results difficult to reproduce.

- It is not described in the paper what kind of model architecture the knowledge selector uses.
- For script data, it is not explained how the authors “retrieve the most relevant scenario” given the query (Is MPNet used here as well to encode the input query and each utterance in the script?).
- It is not described how a word is selected from the dictionary given the query, and how the authors preprocess the Wiktionary to create the dictionary knowledge that would serve as the input, especially when the word contains more than one meaning.
- It is not clear how many knowledge instances are retrieved from the knowledge memory and attached to the query (It seems like one).
- The actual storage footprint of each external knowledge memory is not reported in the paper.
- [Minor point] It is not clearly written in Section 2 whether the authors further train MPNet or use it off-the-shelf (It seems to be the latter case).
- [Minor point] The motivation behind why the authors explored the use of only structured knowledge resources and did not include unstructured knowledge sources as a type of knowledge memory such as Wikipedia, remains unclear.
- [Minor point] The formula for the balancing loss refers to the paper of SwitchTransformer, and is missing in this paper.

**Strength And Weaknesses:**

**Strengths**

- The paper proposes a novel semi-parametric method to use the external memory of heterogeneous types of structured knowledge, which demonstrates its effectiveness across many tasks in various setups with a relatively small number of parameters.

**Weaknesses**

- The paper lacks ablation studies to investigate the effectiveness of each component of the proposed method.
- Several important details of the method are missing, making it less clear and the experimental results difficult to reproduce.

**Suggested Ablation Studies & Analyses**

- In order to see how important is the knowledge-selecting mechanism, it would be interesting to see the performance of KiC without the knowledge selector where all knowledge memories are merged into one and then retrieved (however, it might be difficult if the retrieval mechanism is different for entity and script knowledge types, which is not clearly written in the paper).
- It would be interesting to compare the performance with the cases where unstructured knowledge source(s) is used to augment the input, in order to show the effect of using heterogeneous structured knowledge resources.
- It would be interesting to see the ratio of the knowledge source types selected by the knowledge selector to solve each task. It would demonstrate the effectiveness of using the balancing loss and might provide the readers with further insights.

**Summary Of The Paper:**

This paper proposes a novel semi-parametric language model architecture dubbed Knowledge-in-Context (KiC) which utilizes external knowledge memories of K (K=6 in the experiments) different structured knowledge types (dictionary, commonsense, Entity, event, script, causality) to make the prediction. When a query is given, the knowledge selector classifies the query into (K + 1) classes, where the 0-th class represents that no external knowledge is required for the query. Then, the knowledge memory that corresponds to the class with the highest predicted probability is selected, and the knowledge retrieved from the selected memory is concatenated with the query to serve as the input to the text-to-text model to generate the output. This can be viewed as a mixture-of-experts (MoE) architecture where the parameters of all experts are shared, and each expert has its own external knowledge base. The authors show the effectiveness of the suggested method through extensive experiments on zero-shot setup, in-domain setup, and MMLU tasks, comparing the performance with the simple and state-of-the-art models. The authors also report that KiC seems to show emergent behaviors at the scale of 0.77B.

**Summary Of The Review:**

The paper proposes a novel semi-parametric method to use the external memory of heterogeneous types of structured knowledge, which demonstrates its effectiveness across many tasks in various setups with a relatively small number of parameters. However, many important details of the method and ablation studies are missing in the paper, making the need to update the manuscript essential to make the explanations clear and add several ablation studies and analyses which could provide more insights to the community.

---

> ### Author Response · Authors · 2022-11-18
> **Response to Reviewer 6HCz (1/2)**
>
> **Q1: Add ablation studies to investigate the effectiveness of each component of the proposed method.**
>
> Thanks for the suggestion. In our updated paper, we have added a new Appendix B to include your suggested ablation studies along with others. Specifically, we have added the following baselines: (i) KiC without knowledge, (ii) KiC with an external memory that contains only plain text (English Wikipedia), (iii) KiC without knowledge-selector but retrieving from a mixture of all knowledge categories, (iv) KiC with a task-adaptive selector, and (v) KiC without generalist. The results are reported in Table 12. First of all, it is important to leverage the knowledge-rich memory; when removing the knowledge memory or replacing it with a plain-text memory that consists of English Wikipedia, the performance would degrade greatly. Second, it is also important to use a knowledge selector to first pick a particular category of knowledge and then retrieve the relevant knowledge pieces from it. When we mix all the knowledge categories together with a single retriever, there would be a significant performance drop. The main reason is that different knowledge categories generally require certain pre-filtering strategy during retrieval (see Appendix A.2). Besides, using a single retriever may generate vastly different scores for candidate knowledge pieces from different knowledge categories. For example, ranking scores of script knowledge are generally lower than other knowledge categories due to the relatively long length. Therefore, such knowledge pieces are very unlikely to be selected in a mixed knowledge memory. Furthermore, we also find that the **instance-adaptive** knowledge selector in our KiC model is crucial in achieving good performance. When we replace it with a **task-adaptive** selector, which picks a fixed knowledge category for all instances from the same task based on the task description, the performance is also noticeably worse. Finally, by comparing KiC without generalist to the original KiC, we also observe that there is a noticeable performance drop, which confirms the importance of allowing the model to ignore all external knowledge for some instances.
>
> **Q2: Show the ratio of the knowledge source types selected by the knowledge selector to solve each task.**
>
> Thanks for the suggestion. In Appendix B of our updated paper, we have added a number of plots that illustrate the distributions of the selected knowledge for each task (Figure 4). The results show that most of the knowledge categories are useful for different tasks. And the knowledge selector is able to pick the most helpful knowledge type for solving its current task. For example, in Word-in-Context (WiC) task, the model mostly retrieves from the dictionary knowledge to help it disambiguate different word senses. In StoryCloze task, it relies more heavily on commonsense knowledge to complete the story ending. For MMLU tasks, since they cover a large variety of subjects (i.e., 57 subjects), it is not surprising that it needs more diverse categories of knowledge. In addition, the results further show that the generalist in KiC is also very important as the model would frequently choose it when solving different tasks. It demonstrates the necessity of allowing the model to ignore all external knowledge categories for some instances.
>
> **Q3: Details about the retrieval mechanism for entity, script, and dictionary knowledge types**
>
> In our updated paper, we have added a new Appendix A to provide more details about how we construct different categories of knowledge pieces along with their retrieval modules.
>
> **Q4: Describe what kind of model architecture the knowledge selector uses.**
>
> Thanks for the suggestion. In our updated paper, we have clarified this point in Section 2.3. Specifically, our knowledge selector consists of a transformer encoder, a mean-pooling, and a $(K+1)$-class linear classifier. We apply the same encoder from our T5 backbone model to the input text sequence from a particular task, which generates a sequence of hidden representation vectors. Then, we apply mean-pooling to them to obtain a fixed-dimension vector, which is fed into the $(K+1)$-way linear classifier to generate the logits of selecting different knowledge categories.

---

> > ### Author Response · Authors · 2022-11-20
> > **Response to Reviewer 6HCz (2/2)**
> >
> > **Q5: How many knowledge instances are retrieved from the knowledge memory and attached to the query (It seems like one)?**
> >
> > In our experiment, we retrieve a total of 10 knowledge pieces from the knowledge memory. We attach them to the input and then truncate it to be within the length restriction (e.g., <= 512). In Appendix A.2 of our updated paper, we have included more details to clarify this point.
> >
> > **Q6: The actual storage footprint of each external knowledge memory.**
> >
> > In Appendix A.1 of the updated paper, we have updated Table 5 to list the storage footprint of each external knowledge memory.
> >
> > **Q7: [Minor point] It is not clear (in Section 2) whether the authors further train MPNet or use it off-the-shelf.**
> >
> > We did not further train MPNet in our work. Instead, we use the off-the-shelf MPNet to encode the knowledge pieces in our external knowledge memory.
> >
> > **Q8: [Minor point] The formula for the balancing loss refers to the paper of SwitchTransformer, and is missing in this paper.**
> >
> > We have added the formula for the load balancing loss in Appendix A.2.

---

### Official Review · Reviewer_3xue · 2022-10-25

**Confidence:** 4
**Correctness:** 3
**Technical Novelty And Significance:** 3
**Empirical Novelty And Significance:** 3
**Recommendation:** 6

**Clarity, Quality, Novelty And Reproducibility:**

The work is original enough and provides good information on training general purpose language models that can utilize external knowledge sources.

The writing quality leaves a lot to be desired:
1, There is no listing of the 39 finetuning tasks. Making it difficult to evaluate the generalization ability of the resulting model.
2, The author should include a reference on the "P3 task categorization framework" and the identity of the "P3 tasks".
3, SwithTransformer -> SwitchTransformer,




**Strength And Weaknesses:**

Strength: Thorough evaluation on a wide range of tasks. Providing a broad picture of the effectiveness of the supplied context.

Weakness:
1, The paper is weak on ablations. The author should include more baseline results, in particular the result where the same (initial) model is trained with the same hyperparameters with no non-trivial knowledge sources available. In their language this baseline should have the generalist only. They also did not analyze e.g. how retrieval noise/quality affect the results. Ideally the author should look deeper into how are the retrieval results helping the model to make predictions that are otherwise difficult.

2, A unified retrieval system that works across different knowledge sources and end-to-end trained with the language model would be more  elegant and removes the expert dispatcher and make the method more freely generalizable to expanded knowledge sources without retraining. But this reviewer acknowledge the value of attempting to use out-of-the-box retrieval systems.

3, The paper claims to have experimental evidence that instance-adaptive selection is superior to task-adaptive selection of the context type. But this claim did not seem to be backed up by numbers.

4, The author should comment in the paper on the degradation on tasks such as OpenBookQA and PIQA when going from the no-context finetuned baseline of table 1 to their flagship model in table 4. Some analysis on the win/loss patterns would be valuable.



**Summary Of The Paper:**

The paper constructed framework where a language model is supplied with various types of external context. The context is curated and retrieved, then concatenated with the language model. The retrieval part is frozen and the language model is initialized with LM adapted T5 and trained together with a dispatcher that selects which knowledge source to retrieve from. The model shows improvement over non-context augmented, similarly finetuned models such as T0 on a wide range of tasks.

**Summary Of The Review:**

The paper attempts at a valuable direction of context augmented language models and provided thorough eval for their framework. The authors demonstrated incremental value of retrieved context from various datasources. It is light on ablations and analysis and rough on the edges in terms of presenting all necessary information and baselines. But overall it is a valuable paper. Improvements on the weaknesses of the paper could lead to better review ratings.

---

> ### Author Response · Authors · 2022-11-18
> **Response to Reviewer 3xue**
>
> **Q1: Include the ablation baseline that does not incorporate knowledge (i.e., the generalist-only model) with the same size and hyper-parameters**
>
> In our original paper, we have already included such baselines, which are named as T0-base and T0-large in Table 2 and Table 4 in the paper. That is, these two models are indeed the generalist only models trained using the exact same hyper-parameters as our KiC-base and KiC-large, respectively. Note that the original T0 paper does not provide any T0 models that are in "Base" or "Large" size. Therefore, these two models can also be viewed as our reproduction of T0 of these sizes, and that is why we name them as T0-base and T0-large. We have clarified this point in the caption of Table 2 of the updated paper. In addition, we have also added some additional ablation study results in our Appendix B of the updated paper.
>
> **Q2: Look deeper into how are the retrieval results helping the model to make predictions.**
>
> Thanks for the suggestion. We have added a new Appendix D in our updated paper to include case studies of retrieved knowledge. In particular, we show examples of retrieved knowledge in Table 15. Different knowledge plays critical roles in various tasks. For instance, in the Hellaswag task, the model can predict that a person will mow the lawn because it finds the commonsense knowledge that a "lawn mover'' is used for cutting grass. Similarly, in the WiC task, the model knows that the two "pockets'' are different with the help of a detailed explanation of different synsets of the word "pocket.'' Last but not least, in the Winogrande task, the model can successfully know that burglary is more likely to be investigated because it finds the event knowledge that burglary is often concluded by an investigator.
>
> **Q3: A unified retrieval system that works across different knowledge sources and end-to-end trained with the language model would be more elegant.**
>
> First of all, as the reviewer pointed out, we mainly focus on using a rich set of off-the-shelf knowledge resources and retrieval systems, which we show are of great value in developing a knowledgeable semi-parametric language models.
> Second, different knowledge categories generally have very different characteristics, and may need knowledge-specific processing in order to accurately retrieve them for each instance. For example, to retrieve the entity knowledge, we need to apply entity linking to the query and pre-filter the candidate knowledge pieces before using MIPS search. (See our newly added Appendix A.2 for more details.) For this reason, it is more natural to have a knowledge selector to pick the most relevant knowledge and then retrieve the relevant knowledge piece from it. To demonstrate the importance of knowledge selector, we added a new ablation baseline that mixes all the knowledge categories together and use MIPS to retrieve the knowledge piece from them --- see Appendix B of the updated paper. Ideally, integrating all different kinds of knowledge together would require a universally powerful knowledge representation and retrieval paradigm, which is still an open research problem. Therefore, we leave this as future work.
>
> **Q4: Experiment results to support the advantage of instance-adaptive selection over task-adaptive selection.**
>
> Thanks for the suggestion. In our newly added Appendix B, we have also added a new ablation baseline that only performs task-adaptive knowledge selection. The results show that instance-adaptive knowledge selection (i.e., our KiC model) is superior over the task-adaptive selection.
>
> **Q5: Why are there performance degradations on tasks such as OpenBookQA and PIQA when going from the no-context finetuned baseline of table 1 to their flagship model (KiC) in table 4.**
>
> Different from single task fine-tuning of Table 1, the results marked with * in Table 4 (such as OpenBookQA and PIQA) are **in-domain zero-shot** evaluation, which means that the training data provided by this task are **NOT** used in our multitask training of KiC. However, some similar tasks (such as other multi-choice QA tasks) have been used in training KiC. Therefore, it is not surprising that these tasks experience a performance drop as in-domain zero-shot setting is generally more difficult than directly finetuning on the original data.  In our updated paper, we have clarified this point in the caption of Table 4.
>
> **Q6: Additional clarifications & typos: (i) listing the 39 finetuning tasks; (ii) add a reference on the "P3 task categorization framework" and the identity of the "P3 tasks"; (iii) SwithTransformer -> SwitchTransformer.**
>
> In the updated paper, we have added a new Appendix B to list all 39 tasks with brief descriptions for each task. In addition, we also addressed all the other minor issues pointed out by the reviewer in our updated paper.

---

> > ### Comment · Reviewer_3xue · 2022-11-22
> > **Thanks for adding the ablation studies**
> >
> > The authors added substantial ablation studies and analysis, thus addressing my biggest concerns. Adjusted my rating accordingly.

---

### Decision · Program_Chairs · 2023-01-20

**Decision:**

Accept: notable-top-25%

**Justification For Why Not Higher Score:**

The amount of engineering required may limit the interest and applicability of the work.

**Justification For Why Not Lower Score:**

The method is novel and interesting, and tackles an important problem. The results are strong, and show the proposed method can allow models to outperform others with many more parameters. A good set of ablations have been adde in the revised submission.


**Metareview: Summary, Strengths And Weaknesses:**

The submission shows how to incorporate different kinds of retrieved knowledge into a pre-trained language model, by training a routing mechanism to retrieve from the correct source.

The method is novel and interesting, and tackles an important problem. The results are strong, and show the proposed method can allow models to outperform others with many more parameters. A good set of ablations have been adde in the revised submission.

The main downsides are the method requires some manual engineering for each knowledge type, and there seems to be some sensitivity to the choice of prompts.

Overall this is a strong submission.

**Note From Pc:**

if the above contains the word "oral" or "spotlight" please see: "oral" presentation means -> notable-top-5% and "spotlight" means -> notable-top-25%. As stated in our emails, we are disassociating presentation type from AC recommendations

**Summary Of Ac-Reviewer Meeting:**

n/a